# META-WEIGHTED DIFFUSION MODEL FOR RELIABLE ONLINE SURGICAL PHASE RECOGNITION

## ABSTRACT

Surgical phase recognition has drawn great attention most recently thanks to its potential downstream applications closely related to human life and health. Despite deep network-based models have made significant advancement in capturing discriminative long-term dependency of surgical videos to achieve improved recognition, they seldom account for exploring and modeling uncertainty of surgical videos, which should be crucial for reliable surgical phase recognition. we categorize the sources of uncertainty into two types, imbalanced phase distribution and low-quality image acquisition, which are inevitable in surgical videos. To address this pivot issue, we introduce a meta-weighted diffusion model (MetaDiff) to take full advantages of meta-learning and deep generative model in tackling uncertainty. For uncertainty caused by image quality, we present a classifier-guided diffusion model to produce countable denoised recognition results, making it possible to measure uncertainty using statistical tools for each video frame. For uncertainty caused by phase distribution, we propose a meta-weighted objective function to optimize the classifier-guided diffusion model, making the classification boundary robust against surgical video uncertainty. We demonstrate outstanding ability of our model through comprehensive benchmarks on Cholec80, AutoLaparo, M2Cai16, and CATARACTS. Experimental results reveal that MetaDiff significantly outperforms state-of-the-art methods, separately achieving accuracies of $95.3\%$, $85.8\%$, $92.2\%$, and $85.1\%$ on Cholec80, AutoLaparo, M2Cai16, and CATARACTS.

## 1 INTRODUCTION

Surgical phases recognition aims to identify the representation of high-level surgical stages depicted in surgical videos Jin et al. (2017). This capability holds potential applications for fruitful downstream tasks, such as automatic indexing of surgical video databases Twinanda et al. (2016b), real-time monitoring of surgical procedures Bricon-Souf & Newman (2007), optimizing surgeons schedules Neumuth (2017), evaluating surgeons' proficiency Liu et al. (2021), etc. The primary objective of surgical phase recognition is to predict the category variable $y \in \mathbb{R}^{L \times C}$ given a video frame $x \in \mathbb{R}^{L \times I}$. The process is characterized by the deterministic function $f(x) \in \mathbb{R}^{L \times C}$ that transforms the video frame $x$ into the category variable $y$. To help alert surgeons and support decision-making in real-time during surgery, we do not use the future information within the video frame of $x$, which is also known as online phase recognition Quellec et al. (2014); Dergachyova et al. (2016), which requires us to design the mapping function $f(\cdot)$ carefully without the information leakage.

In recent years, deep neural network-based models He et al. (2016); Vaswani et al. (2017) has shown promising performance in automated surgical phase recognition via designing complex deterministic functions $f(\cdot)$ Jin et al. (2021); Liu et al. (2023b); Ding & Li (2022); Zhao & Krähenbühl (2022); Rivoir et al. (2024). To capture and discover long-term spatial information in surgical videos, Transformer-based methods are proposed Tao et al. (2023); Yue et al. (2023). To enhance the efficiency and effectiveness of Transformer-based methods while avoiding future information leakage , SKiT Liu et al. (2023b) introduces a key pooling operation with a time complexity of $\mathcal{O}(1)$. Most recently, convolution neural network-based models have regained attention due to their awareness of the pitfalls of batch normalization for effective end-to-end learning Rivoir et al. (2024). Recognizing the labor-intensive and time-consuming nature of video annotation in surgical phase recognition, UATD Ding & Li (2022) and VTDC Shi et al. (2021) have been separately proposed to use timestamp annotation and semi-supervised information to alleviate the burden of manual labeling.

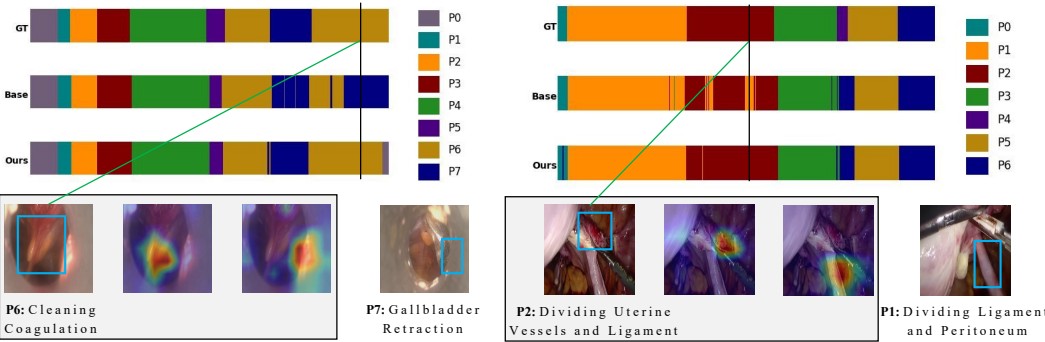

Figure 1: Visualizations are presented on M2Cai16 (left) and AutoLaparo (right) datasets, where the left blue box per dataset indicates the target organ and tool that should be focused on while the right peer represents the incorrectly focused area. Surgical phases are best viewed in color of ribbon diagrams, such as $P_0$ to $P_7$ in M2Cai16. Phase names can be found in Appendix A.3. We find that imbalanced phase distribution and low-quality image acquisition are inevitable in surgical videos, which may cause unexpected errors. To address this, we propose an uncertainty-aware model, called MetaDiff, for reliable online surgical phase recognition.

However, the previous methods neglect the uncertainty nature consisted of phase distribution and image quality in surgical videos as shown in Fig. 1. In laparoscopic rectal cancer surgery, free rectal movements are much more frequent than other movements because they are central to the procedure. Conversely, digestive tract reconstruction action occurs infrequently due to their fixed process, resulting in imbalanced phase distribution. Additionally, irregular changes in camera perspective caused by some emergencies during the surgery process will affect the quality of the surgical video frames. Therefore, overlooking the uncertainty in surgical videos may lead to unforeseen suboptimal outcomes and significantly misguide downstream surgical tasks and pose risks to human health.

To address this pivot issue, in this paper, we propose a novel meta-weighted diffusion model (MetaDiff) for reliable online surgical phase recognition by accurately describing and solving the joint distribution of phase variable $y$ and video frame $x$ in an uncertainty-aware manner. To achieve this, we take full advantages of the deep diffusion model Ho et al. (2020); Song et al. (2020) and meta-learning Guo et al. (2022); Shu et al. (2019) to address the challenges of uncertainty modeling and optimization. On one hand, we introduce a Classifier-guided Diffusion Model (CDM) to quantitatively describe the behaviour of uncertainty caused by video frame quality thanks to stochastic nature of generative models. On the other hand, we present a Meta-weighted Optimization Algorithm (MOA) to model the behavior of uncertainty caused by phase distribution. Notably, the MOA makes optimization via collecting the outputs of CDM, providing an effective approach for making the classification boundary tight even given some low quality video frames. We summarize the main contributions as follows: 1) We are the first one to bring attention to uncertainty nature in surgical videos for reliable online surgical phase recognition. 2) We propose a novel MetaDiff model to implement uncertainty-aware phase recognition by modeling and solving uncertainty with CDM and MOA, respectively. 3) Experiments on four popular benchmarks including Cholec80, AutoLaparo, M2Cai16, and CATARACTS demonstrate that MetaDiff outperforms recently developed competitive baselines across various evaluation metrics, establishing new state-of-the-art (SOTA) performance.

## 2 BACKGROUND

### 2.1 DIFFUSION PROBABILISTIC MODEL

Diffusion Probabilistic models Song et al. (2020) belong to a family of generative models that learn the data distribution based on the Gaussian, typically expressed as $p_{\theta}(\boldsymbol{y}_0) = \int p_{\theta}(\boldsymbol{y}_{0:T}) d\boldsymbol{y}_{1:T}$, where $\{\boldsymbol{y}_t\}_{t=1}^{T}$ are latent variables, $\theta$ is the set of learnable parameters. Denoising Diffusion Probabilistic Model (DDPM) Ho et al. (2020) is one prominent example, which comprises a forward diffusion process alongside a reverse denoising process. During forward diffusion, noise is incrementally introduced, ultimately converting the initial variable $\boldsymbol{y}_0$ into Gaussian noise $\boldsymbol{y}_T$ across $T$ steps:

$$q(\boldsymbol{y}_{1:T}|\boldsymbol{y}_0) = \prod_{t=1}^{T} q(\boldsymbol{y}_t|\boldsymbol{y}_{t-1}), \quad q(\boldsymbol{y}_t|\boldsymbol{y}_{t-1}) = \mathcal{N}(\boldsymbol{y}_t; \sqrt{1-\beta_t}\boldsymbol{y}_{t-1}, \beta_t\mathbf{I}) \quad (1)$$

where $\beta_t$ is the noise level that typically set to a small constant. A notable characteristic of the forward process is that $q(\boldsymbol{y}_t|\boldsymbol{y}_0) = \mathcal{N}(\boldsymbol{y}_t; \sqrt{\alpha_t}\boldsymbol{y}_0, (1 - \alpha_t)\mathbf{I})$, $\alpha_t = \prod_{t=1}^{T}(1 - \beta_t)$. Utilizing a Markov chain with trainable Gaussian transitions, the denoising process from $\boldsymbol{y}_t$ back to $\boldsymbol{y}_0$ unfolds as:

$$p_{\boldsymbol{\theta}}(\boldsymbol{y}_{0:T}) = p_{\boldsymbol{\theta}}(\boldsymbol{y}_T)\prod_{t=1}^{T} p_{\boldsymbol{\theta}}(\boldsymbol{y}_{t-1}|\boldsymbol{y}_t), \quad p_{\boldsymbol{\theta}}(\boldsymbol{y}_{t-1}|\boldsymbol{y}_t) = \mathcal{N}(\boldsymbol{y}_{t-1}; \boldsymbol{\mu}_{\boldsymbol{\theta}}(\boldsymbol{y}_t, t), \sigma_t^2\mathbf{I}) \quad (2)$$

where $\boldsymbol{\mu}_{\boldsymbol{\theta}}(\boldsymbol{y}_t, t) = \frac{1}{\sqrt{\alpha_t}}(\boldsymbol{y}_t - \frac{\beta_t}{\sqrt{1-\alpha_t}}\boldsymbol{\epsilon}_{\boldsymbol{\theta}}(\boldsymbol{y}_t, t))$. Besides, a noise prediction network $\boldsymbol{\epsilon}_{\boldsymbol{\theta}}(\cdot)$ is adopted to minimize the regression loss denoted as $\min_{\boldsymbol{\theta}} \mathbb{E}_{t,\boldsymbol{y}_0,\boldsymbol{\epsilon}\sim\mathcal{N}(\mathbf{0},\mathbf{I})}\|\boldsymbol{\epsilon} - \boldsymbol{\epsilon}_{\boldsymbol{\theta}}(\boldsymbol{y}_t, t)\|_2^2$. With a well trained $\boldsymbol{\epsilon}_{\boldsymbol{\theta}}(\cdot)$, representative latent variables can be systematically generated from random Gaussian noise. Previous researches have substantiated the efficacy of diffusion models across various forms of latent variables, encompassing images Rombach et al. (2022) and multi-time series Shen & Kwok (2023).

## 2.2 LEARNING TO RE-WEIGHT EXAMPLES IN IMBALANCED CLASSIFICATION

Re-weighting the loss function is a widely employed tactic for addressing imbalanced data issue Guo et al. (2022). It treats the weight assigned to each instance as a trainable parameter, enabling the learning of a balanced model for both minority and majority categories through optimization of the weighted loss function. Typically, the optimal weight is optimized on a balanced meta dataset.

$$\boldsymbol{\phi}^*(\boldsymbol{w}) = \arg\min_{\boldsymbol{\phi}} \sum_{i=1}^{N} \boldsymbol{w}^i \mathcal{L}_{train}^i(\boldsymbol{\phi}), \quad \boldsymbol{w}^* = \arg\min_{\boldsymbol{w}} \frac{1}{M}\sum_{j=1}^{M} \mathcal{L}_{meta}^j(\boldsymbol{\phi}^*(\boldsymbol{w})) \quad (3)$$

where $\boldsymbol{w} \in \mathbb{R}^N$ is the weight vector of all training instances, $\boldsymbol{\phi}$ is the set of classifier parameters, $\mathcal{L}_{train}^i$ and $\mathcal{L}_{meta}^j$ are separately the loss functions of pairs $(\boldsymbol{x}^i, \boldsymbol{y}^i)$ and $(\widetilde{\boldsymbol{x}}^j, \widetilde{\boldsymbol{y}}^j)$ from the imbalanced training dataset and the balanced meta dataset, which is downsampled from the training dataset.

## 2.3 PIW AND PAIRED TWO SAMPLE t-TEST FOR ASSESSING UNCERTAINTY

Prediction Interval Width (PIW) is a statistical measure to assess the uncertainty of prediction results. PIW indicates the width of a certain $\tau \cdot 100$ per cent PI, for any value of $\tau \in (0, 1)$:

$$\text{PIW}(\tau) = \frac{1}{n}\sum_{i=1}^{n}(u_i - l_i), \quad u_i = q_{(1+\tau)/2}^i, \quad l_i = q_{(1-\tau)/2}^i \quad (4)$$

where $l_i$ and $u_i$ are the lower and the upper bounding quantiles that together define a $\tau \cdot 100$ per cent PI, $q_{(1-\tau)/2}^i$ and $q_{(1+\tau)/2}^i$ are the $(1 - \tau)/2$ and $(1 + \tau)/2$ quantiles of the predictive distribution of $\boldsymbol{y}^i$. Lower PIW values imply higher sharpness with less uncertainty, which are preferred in practice.

For single-label classification model, the probabilities are dependent between two classes due to the softmax layer, therefore, we use the Paired Two-Sample t-Test (PTST) Fan et al. (2021), which is an inferential statistical test, to determine whether there is a statistically significant difference between the means of two groups. To obtain the PTST, we calculate the difference between paired observations, often referred to as the t-statistic $T = \frac{\overline{Y}}{s/\sqrt{N}}$, where $\overline{Y}$ is the mean difference between paired observations, $s$ stands for the standard deviation of the differences, and $N$ is the number of observations. We use this t-statistic and the t-distribution to determine the corresponding p-value.

## 3 PROPOSED METHOD

In this section, we introduce the Meta-weighted Diffusion model (MetaDiff) for reliable online surgical phase recognition. We begin with a brief overview of MetaDiff, followed by a discussion of its two critical components: Classifier-guided Diffusion Model in Sec. 3.1 and Meta-weighted Optimization Algorithm (MOA) in Sec. 3.2. As shown in Fig. 2, CDM consists of a Spatial-temporal Feature Extractor (SFE), a diffusion process, and a reversed diffusion process. Given surgical videos, the SFE provides rough predictions serving as conditional information to guide the followup reversed diffusion to denoise for obtaining reliable predictions. Notably, each video frame may produce multiple prediction trajectories after denoising, which are modestly collected by the MOA used to learn parameters of CDM for phase recognition in an uncertainty-aware manner.

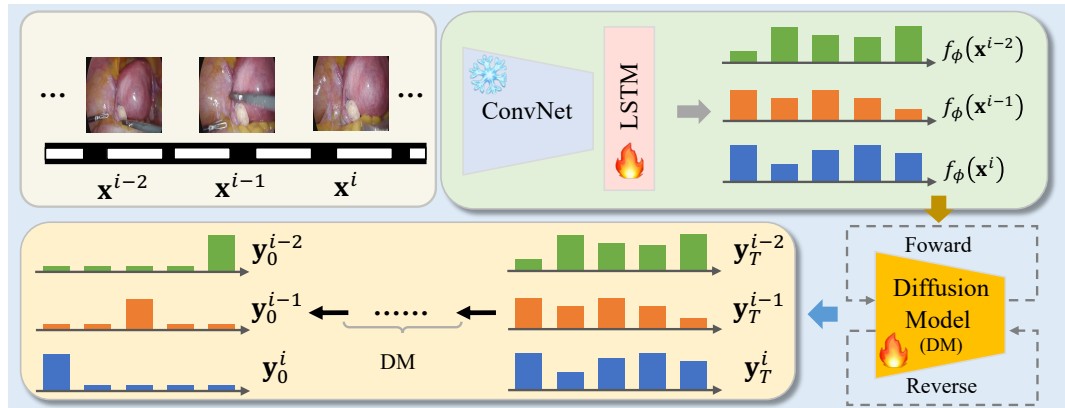

Figure 2: The overview of our proposed MetaDiff, which consists of a Classifier-guided Diffusion Model (CDM) and a Meta-weighted Optimization Algorithm (MOA). Maintaining generality, we employ a simple yet effective backbone $f_\phi$, ConvNext + LSTM, to deliver rough predictions. The upper part illustrates the data flow of obtaining rough predictions, while the lower part shows how CDM obtains clean prediction $\boldsymbol{y}_0^i$ from rough prediction $f_\phi(\boldsymbol{x}^i)$ for the $i$-th frame. The MOA is designed to train the CDM so that the surgical phase recognition could be robust against uncertainty.

## 3.1 CLASSIFIER-GUIDED DIFFUSION MODEL

**Spatial-Temporal Feature Extractor for coarsed predictions.** Existing online surgical phase recognition models Jin et al. (2021); Liu et al. (2023b); Ding & Li (2022); Zhao & Krähenbühl (2022); Rivoir et al. (2024) have primarily focused on learning powerful spatial-temporal representations to achieve robust phase recognition in surgical videos, which can last for several hours and exhibit strong dependencies among different phases. In contrast, we advocate utilizing representations captured by any well-established SFEs rather than designing new SFE architectures. This approach has two distinct advantages: i) we argue that using predictions from well-designed and widely accepted SFEs as conditions for the follow-up diffusion process of CDM is more effective than relying on self-made conditions, as extensive research into the properties of surgical videos has led to the development of specialized SFEs for this purpose; ii) these SFEs demonstrate a strong ability to estimate surgical phase predictions, and using them as conditions for the diffusion may enhance the flexibility of uncertainty estimation, thereby further simplifying the generative process. Given the input video frame $\boldsymbol{x} \in \mathbb{R}^{L \times I}$ and a SFE $f_\phi(\cdot)$, the $j$-th generated prediction $\boldsymbol{z}^{i,j} \in \mathbb{R}^C$ can be expressed as:

$$\boldsymbol{z}^{i,j} \sim \mathcal{N}(\boldsymbol{z}^i; \boldsymbol{\mu}_{\boldsymbol{z}^i}, \sigma \mathbf{I}), \quad \boldsymbol{\mu}_{\boldsymbol{z}^i} = g(f_\phi(\boldsymbol{x}^i)), \quad i = 1, ..., L \tag{5}$$

where we model $g(\cdot)$ as a nonlinear mapping function using a neural network. For simplicity while maintaining generality, we employ a ConvNeXt Liu et al. (2022) cascaded with a LSTM to compute rough predictions from video frames. Importantly, using LSTM for extracting spatial-temporal features is safe for online surgical phase recognition because it avoids utilizing future frames during prediction. To summarize the information captured by the SFE, we define a conditional distribution. The conditional embedding $\boldsymbol{z}^{i,j}$ sampled from this distribution is used for the subsequent diffusion process of CDM. Given the ground truth $\boldsymbol{y}_0^i \in \mathbb{R}^C$ and the conditional embedding $\boldsymbol{z}^{i,j}$, we ensure the representativeness by minimizing a cross entropy loss $\mathcal{L}_{CE}^i(\phi) = -\sum_{c=1}^C \boldsymbol{y}_0^{i,c} log \boldsymbol{z}^{i,j,c}$.

**Forward and Backward Diffusion Process.** Different from the vanilla diffusion models that assume the endpoint of diffusion process to be $\mathcal{N}(\mathbf{0}, \mathbf{I})$, in this work, we model the diffusion process endpoint through the incorporation of the conditional embedding $\boldsymbol{z}^{i,j}$ as $p(\boldsymbol{y}_T^i | \boldsymbol{z}^{i,j}) = \mathcal{N}(\boldsymbol{y}_T^i; \boldsymbol{z}^{i,j}, \mathbf{I})$. With a diffusion schedule $\beta_t \in (0, 1)$ for $t = 1, ..., T$, the forward process is:

$$q(\boldsymbol{y}_t^i | \boldsymbol{y}_{t-1}^i, \boldsymbol{z}^{i,j}) = \mathcal{N}(\boldsymbol{y}_t^i; \sqrt{1 - \beta_t} \boldsymbol{y}_{t-1}^i + (1 - \sqrt{1 - \beta_t}) \boldsymbol{z}^{i,j}, \beta_t \mathbf{I}) \tag{6}$$

Inspired by the DDPM Ho et al. (2020), we can sample $\boldsymbol{y}_t^i$ from $\boldsymbol{y}_0^i$ with an arbitrary timestep $t$ as:

$$q(\boldsymbol{y}_t^i | \boldsymbol{y}_0^i, \boldsymbol{z}^{i,j}) = \mathcal{N}(\boldsymbol{y}_t^i; \sqrt{\alpha_t} \boldsymbol{y}_0^i + (1 - \sqrt{\alpha_t}) \boldsymbol{z}^{i,j}, (1 - \alpha_t) \mathbf{I}) \tag{7}$$

where $\overline{\alpha}_t = 1 - \beta_t$ and $\alpha_t = \prod_t \overline{\alpha}_t$. The mean term in Eq. 7 is an interpolation between the ground truth label encoding $\boldsymbol{y}_0^i$ and the conditional embedding $\boldsymbol{z}^{i,j}$. We gradually add noise to a clean

one-hot encoded label and transform it into a rough prediction vector throughout the forward process. For model training purpose and to facilitate sampling at test time, we derive a tractable backward process posterior corresponding to the forward process in Eq. 6 and Eq. 7 and express it as:

$$
q(\boldsymbol{y}_{t-1}^i|\boldsymbol{y}_t^i, \boldsymbol{y}_0^i, \boldsymbol{z}^{i,j}) = \mathcal{N}(\boldsymbol{y}_{t-1}^i; \gamma_0\boldsymbol{y}_0^i + \gamma_1\boldsymbol{y}_t^i + \gamma_2\boldsymbol{z}^{i,j}, \gamma_3\beta_t\mathbf{I})
$$

$$
\gamma_0 = \frac{\beta_t\sqrt{\alpha_{t-1}}}{1-\alpha_t}, \quad \gamma_1 = \frac{1-\alpha_{t-1}\sqrt{\overline{\alpha}_t}}{1-\alpha_t},
$$

$$
\gamma_2 = 1 + \frac{(\sqrt{\alpha_t}-1)(\sqrt{\overline{\alpha}_t}+\sqrt{\alpha_{t-1}})}{1-\alpha_t}, \quad \gamma_3 = \frac{1-\alpha_{t-1}}{1-\alpha_t}
$$

(8)

The derivation can be found in Appendix A.1. Notably, given a specific conditional embedding $\boldsymbol{z}^{i,j}$, we can employ statistic tools such as PIW and PTST as described in Sec. 2.3 to measure the strength of uncertainty based on multiple trajectories generated through the backward diffusion process. Now the question is how to learn the CDM so that the trajectory regions can be as close as the true label.

## 3.2 META-WEIGHTED OPTIMIZATION ALGORITHM

So far, we have described the behavior of uncertainty caused by surgical video quality. When it comes to train the CDM, uncertainty caused by phase distribution raises, such as imbalanced phase distribution due to the frequency of surgical phases varies greatly. To address these issues, we propose MOA to train CDM for reliable surgical phase recognition. Given a ground truth encoding $\boldsymbol{y}_0^i$ and its conditional embedding $\boldsymbol{z}^{i,j}$ derived from the video frame $\boldsymbol{x}^i$, and intermediate variables $\boldsymbol{y}_{1:T}^i$ generated by the CDM, our goal is to maximize the Evidence Lower BOund (ELBO) written as:

$$
logp_{\boldsymbol{\Theta}}(\boldsymbol{y}_0^i|\boldsymbol{z}^{i,j}) \geq \mathcal{L}_{\text{ELBO}}^i(\boldsymbol{\Theta}) = \mathbb{E}_{q(\boldsymbol{y}_{1:T}^i|\boldsymbol{y}_0^i, \boldsymbol{z}^{i,j})}[log\frac{p_{\boldsymbol{\Theta}}(\boldsymbol{y}_{0:T}^i|\boldsymbol{z}^{i,j})}{q(\boldsymbol{y}_{1:T}^i|\boldsymbol{y}_0^i, \boldsymbol{z}^{i,j})}]
$$

$$
= \mathbb{E}_q[-logp_{\boldsymbol{\Theta}}(\boldsymbol{y}_0^i|\boldsymbol{y}_1^i, \boldsymbol{z}^{i,j})] + \mathbb{E}_q[\text{KL}(q(\boldsymbol{y}_T^i|\boldsymbol{y}_0^i, \boldsymbol{z}^{i,j})||p(\boldsymbol{y}_T^i|\boldsymbol{z}^{i,j}))] \quad (9)
$$

$$
+ \sum_{t=2}^T \mathbb{E}_q[\text{KL}(q(\boldsymbol{y}_{t-1}^i|\boldsymbol{y}_t^i, \boldsymbol{y}_0^i, \boldsymbol{z}^{i,j})||p_{\boldsymbol{\Theta}}(\boldsymbol{y}_{t-1}^i|\boldsymbol{y}_t^i, \boldsymbol{z}^{i,j}))]
$$

where $\boldsymbol{\Theta} = \{\boldsymbol{\phi}, \boldsymbol{\theta}\}$ denotes the learnable parameters of CDM, $\text{KL}(q||p)$ is Kullback-Leibler (KL) divergence from distribution $p$ to $q$. The intermediate objective function can be expressed as:

$$
\mathcal{L}(\boldsymbol{\Theta}) = \frac{1}{N}\sum_{i=1}^N \mathcal{L}^i(\boldsymbol{\Theta}), \quad \mathcal{L}^i = \mathcal{L}_{CE}^i(\boldsymbol{\phi}) + \mathcal{L}_{ELBO}^i(\boldsymbol{\Theta}) \quad (10)
$$

where $N$ is the total number of video frames at training time. However, directly employing $\mathcal{L}$ to optimize $\boldsymbol{\Theta}$ can easily lead to sub-optimal solutions due to the uncertainty caused by imbalanced nature of surgical videos. In other words, the model would be biased towards phases with a majority of video frames and might rise potential risks in human health.

To address this issue, we introduce a meta-learning method for reforming the intermediate loss through reweighting. Inspired by Guo et al. (2022), we firstly construct a meta dataset $\mathcal{D}_{meta} = \{\widetilde{\boldsymbol{x}}^i, \widetilde{\boldsymbol{y}}^i\}_{i=1}^M$ comprising $M$ video frames uniformly downsampled from the original imbalanced training dataset. The remaining video frames excluded from $\mathcal{D}_{meta}$ are defined as new training dataset $\mathcal{D}_{train} = \{\boldsymbol{x}^i, \boldsymbol{y}^i\}_{i=1}^{N-M}$. For simplicities of understanding and notation, we set each video frame only includes a single frame image. Inspired by Shu et al. (2019), we then employ a meta-weight net $h(\cdot; \boldsymbol{w})$ parameterized by $\boldsymbol{w}$ to compute the weight for each video frame from training dataset. After that, the optimization is conducted using the weighted intermediate loss function. It is important to note that the meta-weight net is initially optimized on the meta dataset, thereby guiding the overall model parameter $\boldsymbol{\Theta}$ optimization in a meta-learning manner. The meta-weight net takes the intermediate loss $\mathcal{L}$ on the training dataset as input and outputs an adaptive vector to reweight $\mathcal{L}$ as:

$$
\boldsymbol{\Theta}^*(\boldsymbol{w}) = \arg\min_{\boldsymbol{\Theta}}\mathcal{L}_{train}(\boldsymbol{\Theta}; \boldsymbol{w}) = \frac{1}{N}\sum_{i=1}^{N-M} h(\mathcal{L}_{train}^i(\boldsymbol{\Theta}); \boldsymbol{w})\mathcal{L}_{train}^i(\boldsymbol{\Theta}) \quad (11)
$$

The parameters $\boldsymbol{w}$ in meta-weight net can be optimized using meta-learning strategy and we have:

$$
\boldsymbol{w}^* = \arg\min_{\boldsymbol{w}}\mathcal{L}_{meta}(\boldsymbol{\Theta}^*(\boldsymbol{w})) = \frac{1}{M}\sum_{i=1}^M \mathcal{L}_{meta}^i(\boldsymbol{\Theta}^*(\boldsymbol{w})) \quad (12)
$$

where $\mathcal{L}^i_{train} = \mathcal{L}^i(\boldsymbol{x}^i, \boldsymbol{y}^i; \boldsymbol{\Theta})$ is calculated on training dataset, and the similar for $\mathcal{L}^i_{meta}$. Obviously, the optimal $\boldsymbol{\Theta}^*$ and $\boldsymbol{w}^*$ is calculated using two nested optimization loops. In practice, we adopt an online method to update $\boldsymbol{\Theta}$ and $\boldsymbol{w}$ efficiently through a single optimization loop, detailed as follows:

**Meta training phase:** Since the optimal parameters $\boldsymbol{\Theta}^*$, which should be robust to imbalanced data distribution, depend on the meta-weight net updates, we firstly update the meta-weight net parameters $\boldsymbol{w}$ in a meta training process. Specifically, given a mini-batch of $\{\boldsymbol{x}^i, \boldsymbol{y}^i\}_{i=1}^n$ and $\{\widetilde{\boldsymbol{x}}^i, \widetilde{\boldsymbol{y}}^i\}_{i=1}^m$ separately sampled from the training and meta datasets. Meta-weight net can be updated using:

$$\hat{\boldsymbol{\Theta}}^t(\boldsymbol{w}) = \boldsymbol{\Theta}^t - \frac{\alpha}{n}\sum_{i=1}^n h(\mathcal{L}^i_{train}(\boldsymbol{\Theta}^t); \boldsymbol{w})\nabla_{\boldsymbol{\Theta}}\mathcal{L}^i_{train}(\boldsymbol{\Theta})\Big|_{\boldsymbol{\Theta}^t}$$
$$\boldsymbol{w}^{t+1} = \boldsymbol{w}^t - \frac{\beta}{m}\sum_{i=1}^m \nabla_{\boldsymbol{w}}\mathcal{L}^i_{meta}(\hat{\boldsymbol{\Theta}}^t(\boldsymbol{w}))\Big|_{\boldsymbol{w}^t} \tag{13}$$

where $\alpha$ and $\beta$ now are the step sizes.

**Meta testing phase:** After obtaining the updated $\boldsymbol{w}^{t+1}$, the meta-weight net should be capable of directing attention to rarely observed video frames. Consequently, we use the updated $\boldsymbol{w}^{t+1}$ to improve the parameters in $\boldsymbol{\Theta}$ of our model, which can be expressed as:

$$\boldsymbol{\Theta}^{t+1} = \boldsymbol{\Theta}^t - \frac{\alpha}{n}\sum_{i=1}^n h(\mathcal{L}^i_{train}(\boldsymbol{\Theta}^t); \boldsymbol{w}^{t+1})\nabla_{\boldsymbol{\Theta}}\mathcal{L}^i_{train}(\boldsymbol{\Theta})\Big|_{\boldsymbol{\Theta}^t} \tag{14}$$

The MetaDiff is trained through iterative parameter updates across two meta-learning phases on diverse mini-batch video frames. MetaDiff exhibits robustness in modeling uncertainties of imbalanced phase distribution and low-quality image acquisition, thereby facilitating reliable online surgical phase recognition. We present the pseudocodes in Appendix A.2 of Algorithms 1 and 2

## 4 EXPERIMENT

### 4.1 EXPERIMENTAL SETUP

**Datasets:** Four surgical phase recognition datasets are utilized to extensively evaluate our model, including Cholec80Twinanda et al. (2016b), M2Cai16Twinanda et al. (2016a), AutoLaparoWang et al. (2022), and CATARACTSAl Hajj et al. (2019). Table 1 is basic statistical information of these datasets. Details are depicted in Appendix A.3.

Table 1: Summary of dataset statistics. Tr No. and Te No. separately represent the number of training and testing videos. C No. is phase number.

| Dataset | Duration | fps(f/s) | Tr/Val/Te No. | C No. |
|---|---|---|---|---|
| Cholec80 | 38min26s | 25 | 40/-/40 | 7 |
| MeCai16 | 38min25s | 25 | 27/-/14 | 8 |
| AutoLaparo | 66min07s | 25 | 10/4/7 | 7 |
| CATARACTS | 10min56s | 30 | 25/-/25 | 19 |

**Evaluation metrics:** We use four widely used metrics including accuracy (Acc), precision (Pr), recall (Re), and Jaccard (Ja) to evaluate the online surgical phase recognition performance. We leverage Prediction Interval Width (PIW) and Paired Two Samples t-Test (PTST) to quantify the model's uncertainty, please refer to Sec. 2.3 for more details. Due to the subjective nature of manual labeling in surgical videos and the ambiguous boundaries between adjacent surgical stages which are noted by Gao et al. (2021); Jin et al. (2021); Yi et al. (2022), Cholec80 and M2Cai16 datasets adopt lenient boundary metrics to access model performance. Specifically, frames predicted belonging to adjacent stages within a 10 seconds window before and after a phase transition are also deemed correct.

**Baselines:** We compare our model, MetaDiff, with most recently proposed state-of-the-art competitors such as PitBN Rivoir et al. (2024), SKiTLiu et al. (2023b), CMTNet Yue et al. (2023), LAST Tao et al. (2023), TMRNet Jin et al. (2021), Trans-SVNet Gao et al. (2021), TeCNO Czempiel et al. (2020), SV-RCNet Jin et al. (2017) and so on, using the metrics introduced above. The results are derived from their respective papers or reproduced using their available official codes.

**Implementation details:** We utilize ConvNeXt Liu et al. (2022) pretrained on ImageNet-1K Krizhevsky et al. (2017) to extract spatial features from videos, followed by LSTM for temporal feature fusion. During training, we freeze the earlier blocks of ConvNeXt and updated only the parameters of its last block. To generate meaningful conditional embeddings for opimizing CDM, we

Table 2: The results (%) of MetaDiff V.S. other competitors on Cholec80 and CATARACTS datasets. The best results are marked in bold.

Table 3: The results (%) of MetaDiff V.S. other competitors on AutoLaparo and M2Cai16 datasets. The best results are marked in bold.

| Dataset | Methods | R | Acc | Pr | Re | Ja |
|---|---|---|---|---|---|---|
| CATARACTS | TransSVNet | | 77.8 | 61.3 | 55.0 | 43.8 |
| | 3DCNN | | 80.1 | 66.2 | 55.7 | 45.9 |
| | SV-RCNet | | 81.3 | 66.0 | 57.0 | 47.2 |
| | PitBN | | 83.3 | 66.8 | 61.8 | 50.3 |
| | DualPyramid | | 84.2 | 69.3 | 66.4 | 53.7 |
| | MetaDiff (Ours) | | **85.1** | **72.1** | **66.6** | **54.2** |
| Cholec80 | Trans-SVNet | ✓ | 90.3 ±7.1 | 90.7 | 88.8 | 79.3 |
| | TeSTra | | 90.1 ±6.6 | 82.8 | 83.8 | 71.6 |
| | Dual Pyramid | | 91.4 | 85.4 | 86.3 | 75.4 |
| | OperA | ✓ | 90.2 ±6.1 | 84.2 | 85.5 | 73.0 |
| | CMTNet | ✓ | 92.9 ±5.9 | 90.1 | 92.0 | 81.5 |
| | LAST | ✓ | 93.1 ±4.7 | 89.3 | 90.1 | 81.1 |
| | LoViT | ✓ | 92.4 ±6.3 | 89.9 | 90.6 | 81.2 |
| | SKiT | ✓ | 93.4 ±5.2 | 90.9 | 91.8 | 82.6 |
| | PitBN | ✓ | 93.5 ±6.5 | 90.0 | 91.9 | 82.9 |
| | MetaDiff (Ours) | | 94.2 ±4.3 | 89.6 | 90.0 | 81.7 |
| | MetaDiff (Ours) | ✓ | **95.3 ±4.1** | **92.9** | **93.1** | **86.0** |

| Dataset | Methods | R | Acc | Pr | Re | Ja |
|---|---|---|---|---|---|---|
| AutoLaparo | SV-RCNet | | 75.6 | 64.0 | 59.7 | 47.2 |
| | TeCNO | | 77.3 | 66.9 | 64.6 | 50.7 |
| | TMRNet | | 78.2 | 66.0 | 61.5 | 49.6 |
| | Trans-SVNet | | 78.3 | 64.2 | 62.1 | 50.7 |
| | LoViT | | 81.4±7.6 | **85.1** | 65.9 | 56.0 |
| | SKiT | | 82.9 ±6.8 | 81.8 | 70.1 | 59.9 |
| | PitBN | | 83.7 ±6.6 | 79.5 | 67.7 | 58.8 |
| | MetaDiff (Ours) | | **85.8 ±6.0** | 82.3 | **71.1** | **61.2** |
| M2Cai16 | SV-RCNet | ✓ | 81.7 ±8.1 | 81.0 | 81.6 | 65.4 |
| | OHFM | ✓ | 85.2 ±7.5 | – | – | 68.8 |
| | TMRNet | ✓ | 87.0 ±8.6 | 87.8 | 88.4 | 75.1 |
| | Not-E2E | ✓ | 84.1 ±9.6 | – | 88.3 | 69.8 |
| | Trans-SVNet | ✓ | 87.2 ±9.3 | 88.0 | 87.5 | 74.7 |
| | CMTNet | ✓ | 88.2 ±9.2 | 88.3 | 88.7 | 76.1 |
| | LAST | ✓ | 91.5 ±5.6 | 86.3 | 88.7 | 77.8 |
| | PitBN | ✓ | 91.1 ±7.2 | 90.0 | 92.5 | 81.4 |
| | MetaDiff (Ours) | ✓ | **92.2 ±5.1** | **91.5** | **92.7** | **82.9** |

initially pretrain the SFE, comprising ConvNeXt and LSTM, using standard cross-entropy loss on an imbalanced training dataset. The feature vectors extracted by ConvNeXt have a dimensionality of 768. Both the LSTM output dimension and the CDM input dimension are set to 512. We employ AdamW Kingma & Ba (2014) to optimize our model, with separate learning rates of 1e-5 for $\Theta$ and 1e-3 for $w$, without weight decay. To ensure fair comparisons, we maintain batch size of 1 and the time window length of 256, consistent with other competitors. All experiments are conducted on a single NVIDIA A100 80GB PCIe GPU. More details can be found in Appendix A.4.

## 4.2 MAIN RESULTS

### 4.2.1 QUANTITATIVE RESULTS AND ANALYSIS

**Online surgical phase recognition:** We conduct comprehensive studies comparing MetaDiff with other state-of-the-art methods for surgical phase recognition on Cholec80, AutoLaparo, M2Cai16, and CATARACTS datasets. Quantitative results for these datasets are separately reported in Table 2 and Table 3. MetaDiff significantly outperforms most of competitors, such as SKiT and LAST, across various metrics including accuracy (Acc), precision (Pr), recall (Re), and Jaccard (Ja). For example, MetaDiff shows improvements on Cholec80 with increase of 1.8% in Acc, 2% in Pr, 1.2% in Re, and 3.1% in Ja compared to the second-best method. Additionally, MetaDiff delivers superior results in Pr, Re, and Ja, effectively addressing imbalanced effects. MetaDiff also achieves lower standard deviations of Acc, with reductions of 0.6%, 0.8%, and 0.5% on Cholec80, AutoLaparo, and M2Cai16 datasets, respectively, compared to the second-best method. Unlike complex architectures such as Transformer-based models used in LoViT and SKiT, MetaDiff employs the simple ConvNeXt+LSTM architecture. We attribute these notable improvements to effectiveness of MetaDiff in addressing the challenges posed by imbalanced and uncertain problems in surgical videos.

**Uncertainty estimation:** We present the results of MetaDiff on uncertainty estimation for evaluating the instance-level prediction confidence under the scope of the entire Cholec80 test dataset in Table 4. Specifically, for each test frame, we generate 100 predictions through the reverse diffusion process, resulting in a $100 \times 7$ matrix. We then compute PIW and PTST based on this matrix. After obtaining the PIW and the PTST from each test frame, we divide the test dataset into two groups by the correctness of majority-vote predictions. We calculate the average PIW of the true phase within each group. We also split the test instances by t-test rejection status, and compute the mean accuracy in each group. For details, please refer to Appendix A.5. As we can see that the mean PIW of the ground truth label among the correct predictions is ($10\times$) narrower than that of the incorrect predictions, indicating that MetaDiff can make correct predictions with much smaller variations. Furthermore, when comparing the mean PIWs across different phases, we observe that the phase indexed as 0 has the lowest accuracy at 39.0% and its incorrect prediction interval is much smaller than other phases. All these evidences suggest that the uncertainties of phase 0 could be especially significant. Moreover, we observe that the accuracy of test instances rejected by the t-Test is significantly higher than that of the not-rejected ones, both across the entire test dataset and within each phase. We point out that these metrics reflect confidence of MetaDiff in the correctness of predictions and have the potential

Table 4: PIW ($\times$ 100) and t-test on Cholec80 dataset.

| Class | Accuracy | PIW | | Acc by t-Test | |
|-------|----------|-----|-----|------|------|
| | | Correct | Incorrect | Reject | Not-Reject (count) |
| all | 81.3% | 0.65 | 13.40 | 91.2% | 50.8%(134) |
| 0 | 39.0% | 0.43 | 1.45 | 39.0% | 20.0%(5) |
| 1 | 97.4% | 0.13 | 10.69 | 97.4% | 60.0%(10) |
| 2 | 83.4% | 0.51 | 8.12 | 83.5% | 37.5%(8) |
| 3 | 96.5% | 0.39 | 17.10 | 96.5% | 52.4%(21) |
| 4 | 88.8% | 0.97 | 27.31 | 88.9% | 50.0%(8) |
| 5 | 78.9% | 3.07 | 23.99 | 79.2% | 43.8%(48) |
| 6 | 84.6% | 3.24 | 45.89 | 84.8% | 64.7%(34) |

Table 5: Complexity and running time analysis on Cholec80 dataset.

| Methods | Params (M) | time (ms) | GFLOPs | Acc (%) |
|---------|-----------|-----------|--------|---------|
| TeCNO | 24.69 | 19 | 4.11 | 88.6 ±7.8 |
| TMRNet | 63.02 | 26 | 8.29 | 90.1 ±7.6 |
| TransSVNet | 24.72 | 19 | 4.15 | 90.3 ±7.1 |
| NotE2E | 22.73 | 49 | 5.72 | 91.5 ±7.1 |
| CMTNet | 26.63 | 33 | 5.56 | 92.9 ±5.9 |
| LAST | 117.26 | 86 | 15.49 | 93.1 ±4.7 |
| MetaDiff-10 | 21.44 | 9 | 5.81 | 95.0 ±4.3 |
| MetaDiff-100 | 21.44 | 76 | 11.96 | 95.3 ±4.1 |
| MetaDiff-500 | 21.44 | 367 | 39.27 | 95.5 ±4.1 |

Table 6: The results (%) V.S. different components of MetaDiff on Cholec80 dataset.

| CDM | Meta | Acc | Pr | Re | Ja |
|-----|------|-----|-----|-----|-----|
| | | 93.5 ±6.5 | 90.0 | 91.9 | 82.9 |
| ✓ | | 94.2 ±5.2 | 91.1 | 92.1 | 83.4 |
| | ✓ | 94.5 ±4.8 | 92.3 | 92.7 | 85.2 |
| ✓ | ✓ | 95.3 ±4.1 | 92.9 | 93.1 | 86.0 |

Table 7: The results (%) on the scope of optimized parameters under Cholec80 dataset.

| Range | Acc | Pr | Re | Ja |
|-------|-----|-----|-----|-----|
| C | 93.4 ±5.2 | 90.7 | 90.7 | 82.7 |
| LSTM+C | 94.2 ±4.5 | 92.0 | 91.0 | 83.0 |
| ConvNeXt#+LSTM+C | 95.3 ±4.1 | 92.9 | 93.1 | 86.0 |
| ConvNeXt+LSTM+C | 90.6 ±5.6 | 88.7 | 89.6 | 81.9 |

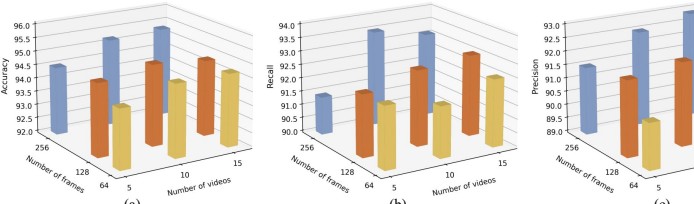 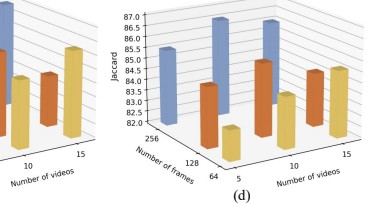

Figure 3: The results (%) of effects on the scale of meta dataset including number of videos and frames when constructing the meta dataset on Cholec80 dataset.

to be applied in mitigating risks during surgical evaluation. Such uncertainty estimation can be used to decide whether to accept the prediction or to refer the instance to experts for further evaluation.

**Ablation study:** We verify the effects of different components on our proposed MetaDiff, results are reported in Table 6. CDM represents whether using conditional diffusion model to denoise from rough predictions. Meta refers to whether employing meta-weight net to reweight the intermediate loss function. On one hand, either equipping CDM or Meta can consistently improve recognition performances across all metrics. On the other hand, combining both CDM and Meta altogether can further boost recognition performances. We are surprised to observe that equipping CDM can improve metrics that reflect imbalance issue (Pr,Re,Ja). It implies that CDM has capability to ameliorate robustness on imbalanced surgical videos thanks to its by-products brought by uncertainty estimation.

**Analysis on hyper-parameters: (1)** As mentioned in the implementation details of Sec. 4.1 that we initially pretrain MetaDiff using standard cross-entropy loss function to obtain meaningful conditional embeddings. Therefore, we investigate the performances V.S. the scope of optimized parameters during fine-tuning and report results in Table 7, where C represents we only fine-tune the classifier parameters. LSTM+C refers to that we fine-tune parameters of both the LSTM and classifier. ConvNeXt#+LSTM+C denotes that we fine-tune the parameters of the last block in ConvNeXt, LSTM, and classifier. ConvNeXt+LSTM+C is that we fine-tune the whole parameters of ConvNeXt, LSTM, and classifier. We find that fine-tuning parameters with proper amount is beneficial for accelerating performances, which might be able to attribute to the fact that basic and general representations of ConvNeXt is essential for robust predictions of MetaDiff. **(2)** We study performances V.S. the scale of meta dataset and report results in Fig. 3. We observe that our model achieves consistent recognition performances even with a small scale meta dataset, demonstrating its significance and suitability for practical online surgical phase recognition applications.

**Confusion matrices:** We visualize the confusion matrices of baseline model (ConvNeXt+LSTM) optimized with standard cross entropy loss and MetaDIff, results are shown in Fig. 5 (b) and (c). Our model ameliorates the performances on minority phases, P5 (phase of Cleaning Coagulation) for instance, the stochastic of which can be found in Table 8.

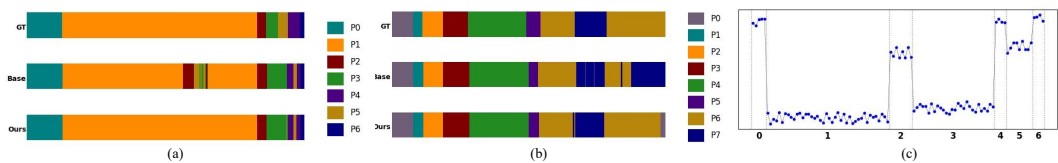

Figure 4: (a) and (b) are ribbon diagrams of ground truth labels, baseline method, and MetaDiff from the top to the bottom under Cholec80 and M2Cai16 datasets. (c) Learned weight vectors on Cholec80 dataset, where x-axis is samples from the current mini-batch and we only mark their labels for clarity.

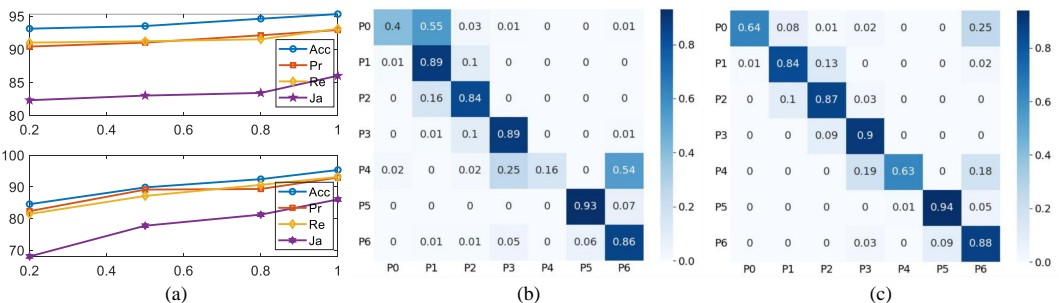

Figure 5: (a) Recognition performances change with the number of training frames in Cholec80. Top (a) are results of ignoring training labels. Bottom (a) are results of ignoring training frames. (b) and (c) are confusion matrices of base model (ConvNext+LSTM) and MetaDiff on AutoLaparo.

**Complexity analysis:** During evaluation, we select the diffusion timestep to $T = 1000$. To accelerate prediction speed, we employ the DDIM Song et al. (2021) sampling strategy, reducing the total sampling requirement effectively to $T = 100$. On one hand, we conduct comparative experiments using different diffusion timesteps and depict results in Table 5. On the other hand, we also compare the complexity of MetaDiff with other competitors. Overall, MetaDiff achieves a satisfactory balance between performance and real-time efficiency.

**Surgical phase recognition in low-data regime:** In practice, annotating surgical videos is labor-intensive and time-consuming, therefore, verifying the effectiveness of our model under low data regime is also crucial. We conduct experiments under two data-limited scenarios, and report the results in Fig. 5(a). Our model also achieves robust performance when the training frames are limited.

### 4.2.2 QUALITATIVE RESULTS AND ANALYSIS

**(1)** We employ ConvNeXt + LSTM optimized with standard cross entropy loss function as baseline model. And we compare ribbon diagrams among ground truth labels, baseline model, and MetaDiff to show the capability of our model in reliable online surgical phase recognition, as shown in Fig. 4 (a) and (b). Taking predicted ribbon diagram on Cholec80 dataset for example, the baseline model easily misclassifies P1 (CalotTriangleDissection) into P2 (framepingCutting) at middle of the video. In contrast, our model effectively avoid such errors. **(2)** We visualize the learned weight vectors of 100 training frames uniformly sampled from each phase and show result in Fig. 4 (c). We find that the learned frame weights for minority phases are typically more prominent than those for majority phases, prompting the model to focus more on frames from minority phases and thereby reducing the risk of overfitting to majority phases. This observation is also consistent with human intuition.

## 5 CONCLUSION

We propose MetaDiff to address a crucial issue that have long been overlooked in reliable online surgical phase recognition, namely uncertainty caused by phase distribution and video quality. MetaDiff employs the CDM to deliver clean predictions, being less disturbed by uncertainty, from rough ones provided by a pre-trained SFE. Additionally, we employ the MOA to train the model so that enabling the CDM to have such uncertainty-aware capability. Our empirical results demonstrate that the proposed model outperforms other specially designed competitors in complex online surgical phase recognition task.

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

## A APPENDIX

### A.1 DERIVATION FOR FORWARD PROCESS POSTERIOR

We derive the forward process posterior in Eq. 8 as follows:

$$
\begin{aligned}
q(\boldsymbol{y}_{t-1}^i|\boldsymbol{y}_t^i, \boldsymbol{y}_0^i, \boldsymbol{z}^{i,j}) &\propto q(\boldsymbol{y}_t^i|\boldsymbol{y}_{t-1}^i, \boldsymbol{z}^{i,j}) q(\boldsymbol{y}_{t-1}^i|\boldsymbol{y}_0^i, \boldsymbol{z}^{i,j}) \\
&\propto \exp\{-\frac{1}{2}[\frac{(\boldsymbol{y}_t^i - (1-\sqrt{\overline{\alpha}_t})\boldsymbol{z}^{i,j} - \sqrt{\overline{\alpha}_t}\boldsymbol{y}_{t-1}^i)^2}{\beta_t} \\
&\qquad + \frac{(\boldsymbol{y}_{t-1}^i - \sqrt{\alpha_{t-1}}\boldsymbol{y}_0^i - (1-\sqrt{\alpha_{t-1}})\boldsymbol{z}^{i,j})^2}{1-\alpha_{t-1}}]\} \\
&\propto \exp\{-\frac{1}{2}[\boldsymbol{A}(\boldsymbol{y}_{t-1}^i)^2 - 2\boldsymbol{B}\boldsymbol{y}_{t-1}^i]\}
\end{aligned}
\tag{15}
$$

where
$$
\boldsymbol{A} = \frac{1-\alpha_t}{\beta_t(1-\alpha_{t-1})}
$$
$$
\boldsymbol{B} = \frac{\sqrt{\alpha_{t-1}}}{1-\alpha_{t-1}}\boldsymbol{y}_0^i + \frac{\sqrt{\overline{\alpha}_{t-1}}}{\beta_t}\boldsymbol{y}_t^i + (\frac{\sqrt{\overline{\alpha}_t}(\overline{\alpha}_t - 1)}{\beta_t} + \frac{1-\sqrt{\alpha_{t-1}}}{1-\alpha_{t-1}})\boldsymbol{z}^{i,j}
$$

According to properties of Gaussian distribution in Eq. 10.100 and Eq. 10.101 of Bishop (2006), the variance of posterior can be expressed as $\frac{1-\alpha_{t-1}}{1-\alpha_t}\beta_t$, and we have $\gamma_3 = \frac{1-\alpha_{t-1}}{1-\alpha_t}$. Meanwhile, the mean of posterior can be written as:

$$
\widetilde{\boldsymbol{\mu}}(\boldsymbol{y}_t^i, \boldsymbol{y}_0^i, \boldsymbol{z}^{i,j}) = \frac{\beta_t\sqrt{\alpha_{t-1}}}{1-\alpha_t}\boldsymbol{y}_0^i + \frac{1-\alpha_{t-1}\sqrt{\overline{\alpha}_t}}{1-\alpha_t}\boldsymbol{y}_t^i + (1 + \frac{(\sqrt{\alpha_t}-1)(\sqrt{\overline{\alpha}_t} + \sqrt{\alpha_{t-1}})}{1-\alpha_t})\boldsymbol{z}^{i,j} \tag{16}
$$

For the sake of simplicity, we define $\widetilde{\boldsymbol{\mu}} = \gamma_0\boldsymbol{y}_0^i + \gamma_1\boldsymbol{y}_t^i + \gamma_2\boldsymbol{z}^{i,j}$ and have:

$$
\gamma_0 = \frac{\beta_t\sqrt{\alpha_{t-1}}}{1-\alpha_t}, \quad \gamma_1 = \frac{1-\alpha_{t-1}\sqrt{\overline{\alpha}_t}}{1-\alpha_t}, \quad \gamma_2 = 1 + \frac{(\sqrt{\alpha_t}-1)(\sqrt{\overline{\alpha}_t} + \sqrt{\alpha_{t-1}})}{1-\alpha_t} \tag{17}
$$

### A.2 PSEUDO CODES FOR TRAINING AND INFERENCE

| **Algorithm 1** Training of MetaDiff |
| --- |
| 1: Initialize parameters $\boldsymbol{\Theta}$ |
| 2: **repeat** |
| 3:   Draw mini-batch from $\mathcal{D}_{train}$ and $\mathcal{D}_{meta}$ |
| 4:   Draw $t \sim \text{Uniform}(1, T)$ |
| 5:   Draw $\epsilon \in \mathcal{N}(\mathbf{0}, \mathbf{I})$ |
| 6:   Draw $\boldsymbol{z}^{\cdot\cdot}$ from Eq. 5 |
| 7:   Compute the loss in Eq. 10 |
| 8:   Update parameters using Eq. 13 and 14 |
| 9: **until** convergence |

| **Algorithm 2** Inference of MetaDiff |
| --- |
| 1: Draw $\boldsymbol{x}^i$ from test dataset |
| 2: Draw $\boldsymbol{z}^{i,j}$ from Eq. 5 |
| 3: **for** $t = T$ to 1 **do** |
| 4:   Calculate: $\hat{\boldsymbol{y}}_0^i = \frac{1}{\alpha_t}(\boldsymbol{y}_t^i - (1-\sqrt{\alpha_t})\boldsymbol{z}^{i,j} - \sqrt{1-\alpha_t}\epsilon_{\boldsymbol{\theta}}(\boldsymbol{y}_t^i, \boldsymbol{z}^{i,j}, t))$ |
| 5:   **if** $t \geq 1$: Draw $\epsilon \sim \mathcal{N}(\mathbf{0}, \mathbf{I})$ |
| 6:     $\boldsymbol{y}_{t-1}^i = \gamma_0\hat{\boldsymbol{y}}_0^i + \gamma_1\boldsymbol{y}_t^i + \gamma_2\boldsymbol{z}^{i,j} + \sqrt{\gamma_3\beta_t}\epsilon$ |
| 7:   **else:** $\boldsymbol{y}_{t-1}^i = \hat{\boldsymbol{y}}_0^i$ |
| 8: **end for** |

### A.3 MORE DETAILS ON DATASETS

Cholec80 Twinanda et al. (2016b) comprises 80 laparoscopic surgical videos, with 7 defined phases annotated by experienced surgeons and an average duration of 39 minutes at 25 fps with resolution either 1920 ×1080 or 854 × 480. We split the dataset into the 40 videos for training and the rest for testing follow Jin et al. (2017). M2Cai16 Twinanda et al. (2016a) consists of 41 laparoscopic surgical videos with resolution 1920×1080 that are segmented into 8 phases by expert physicians. Following Yi & Jiang (2019), we split the dataset into the 27 videos for training and the 14 for testing. AutoLaparo Wang et al. (2022) includes 21 laparoscopic hysterectomy videos, with 7 phases annotated by experienced surgeons and an average video duration of 66 minutes recorded at 25 fps with resolution 1920 ×1080. FollowingLiu et al. (2023a), we split the dataset into 10 videos for training, and 7 videos for testing. CATARACTS Al Hajj et al. (2019) comprises 50 videos of

Table 8: Phase names of Cholec80 dataset.

| Phase | P0 | P1 | P2 | P3 | P4 | P5 | P6 |
|---|---|---|---|---|---|---|---|
| Name | Preparation | CalotTriangle Dissection | framepingCutting | Gallbladder Dissection | Gallbladder Packaging | Cleaning Coagulation | Gallbladder Retraction |
| Num | 3727 | 36877 | 7329 | 24119 | 5716 | 5222 | 3314 |

Table 9: Phase names of M2Cai16 dataset.

| Phase | P0 | P1 | P2 | P3 | P4 | P5 | P6 | P7 |
|---|---|---|---|---|---|---|---|---|
| Name | TrocarPlacement | Preparation | CalotTriangle Dissection | framepingCutting | Gallbladder Dissection | Gallbladder Packaging | Cleaning Coagulation | Gallbladder Retraction |
| Num | 4913 | 2763 | 17062 | 7607 | 16850 | 1847 | 8524 | 7989 |

Table 10: Phase names of AutoLaparo dataset.

| Phase | P0 | P1 | P2 | P3 | P4 | P5 | P6 |
|---|---|---|---|---|---|---|---|
| Name | Preparation | Dividing Ligament and Peritoneum | Dividing Uterine Vessels and Ligament | Transecting the Vagina | Specimen Removal | Suturing | Washing |
| Num | 739 | 12957 | 9841 | 5814 | 296 | 6808 | 3756 |

Table 11: Selected phase number of CATARACTS dataset.

| Phase | P0 | P1 | P2 | P3 | P4 | P5 | P6 | P7 | P8 | P9 | P10 | P11 | P12 | P13 | P14 | P15 |
|---|---|---|---|---|---|---|---|---|---|---|---|---|---|---|---|---|
| Num | 22338 | 150 | 473 | 3074 | 2055 | 5026 | 1736 | 5270 | 8173 | 2310 | 8179 | 2039 | 1844 | 1595 | 4522 | 5184 |

cataract surgeries, with an average duration of 10 minutes and 56 seconds per video. Each video has a frame rate of 30 FPS and a resolution of $1920 \times 1080$ pixels. The dataset includes 19 stages to be identified and is split into 25 training sets and 25 test sets following Al Hajj et al. (2019). All videos are subsampled to 1 fps following Twinanda et al. (2016b), and frames are resized into $250 \times 250$. We separately illustrate phase names of the four surgical video datasets in Table 8, Table 9, Table 10 and Table 11, the stage name of the CATARACTS is not explicitly given, only the number of corresponding stage frames is displayed. Due to limited space, in Table 11, we only give part of phase number in CATARACTS dataset. In addition, we briefly illustrate the differences between the four surgical video datasets using video frames in Fig. 6.

## A.4 MORE DETAILS ON ARCHITECTURES

In our experiments, we set the number of timesteps as $T = 100$ and employed a linear noise schedule with $\beta_1 = $1e-4 and $\beta_t = 0.02$. The conditional embedding $z^{..}$ are configured to be 512 dimension. For LSTM in SFE, we use a two layers LSTM, each layer has 512 hidden units. For architecture of CDM, we initially use a linear embedding for the timestep. We then concatenate $y_t^i$ and $y_0^i$ and feed them into a three layers MLP, each with an output dimension of 512. We conduct Hadamard product between the output vector and the corresponding timestep embedding, followed by a Softplus non-linear function. Finally, we use another fully-connected layer activated with Softmax function to map the vector to the rough predictions, also known as conditional embeddings in our work.

## A.5 MORE DETAILS ON PIW AND PTST

Prediction Interval Width (PIW) and Paired Two-Sample t-Test (PTST) are used to assess the predictive uncertainty of our proposed MetaDiff. We provide the mean PIW among correct and incorrect predictions, and the mean accuracy among instances rejected and not-rejected by the PTST for all test instances. And we describe details of their calculation process as follows.

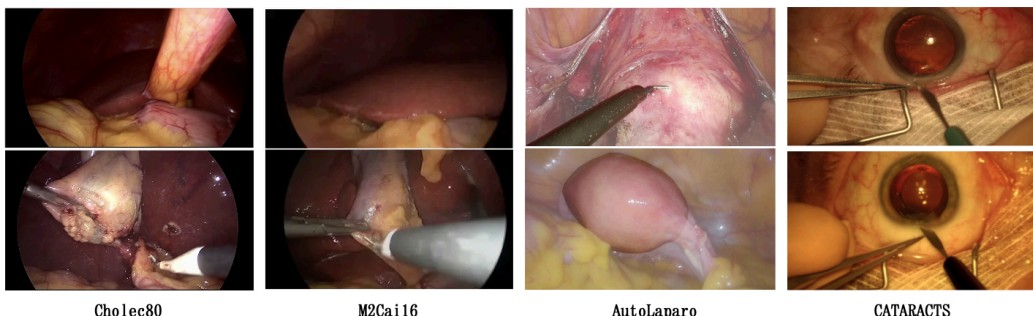

Figure 6: Showcases of surgical video frame for the four datasets.

For ease of representation, we set $n_{test}$ as the number of instances in the test dataset and $n_{sample}$ as the number of samples taken for uncertainty assessment for each instance. The data for all instances in the test dataset is represented as $\boldsymbol{x} \in \mathbb{R}^{n_{test} \times I}$, where $I$ is feature dimension. Their corresponding ground-truth label with one-hot encoding is denoted as $\boldsymbol{y} \in \mathbb{R}^{n_{test} \times C}$, where $C$ is phase number. When using MetaDiff for inference on the test dataset, each input instance is used to generated $n_{sample}$ denoised outputs (predicted vectors) $\hat{\boldsymbol{y}} \in \mathbb{R}^{n_{test} \times n_{sample} \times C}$ by reverse diffusion. By taking the maximum predicted probability among the $n_{sample}$ predictions for each instance, we obtain the predicted recognition results $\overline{\boldsymbol{y}} \in \mathbb{R}^{n_{test} \times C}$ for all instances. Additionally, extracting the top two maximum probabilities from the $n_{sample}$ predictions yields two data distributions, each containing $n_s ample$ values for each instance. These two distributions serve as the two input samples for PTST statistical testing, denoted collectively as $\boldsymbol{t} \in \mathbb{R}^{n_{test} \times n_{sample} \times 2}$. Comparing the predicted results $\overline{\boldsymbol{y}}$ with the ground-truth label $\boldsymbol{y}$ allows us to determine the number of correctly classified and misclassified samples, thereby the recognition accuracy is obtained. Subsequently, we compute the PIW values between the 2.5th and 97.5th percentiles of the predicted probabilities separately for correct and incorrect samples, averaging these values to derive the PIW for correct and incorrect samples. Besides, we conduct hypothesis tests on the obtained dual-sample distribution $\boldsymbol{t}$, yielding a p-value for each instance, which is compared against the given significance level $\rho = 0.05$ to classify instances as 'reject' or 'no-reject' groups. It's noteworthy that these hypothesis tests are based on the largest and second largest predicted probabilities for each instance. Therefore, the hypothesis is that "the top two maximum predicted values are the same". Larger differences between these probabilities indicate higher accuracy in the maximum predicted probability, identifying 'reject' samples as those with better predictive performance. Finally, we tally the number of correctly predicted samples in the 'reject' and 'no-reject' groups, calculating the corresponding accuracy for each group.

## A.6 REBUTTAL RESULTS

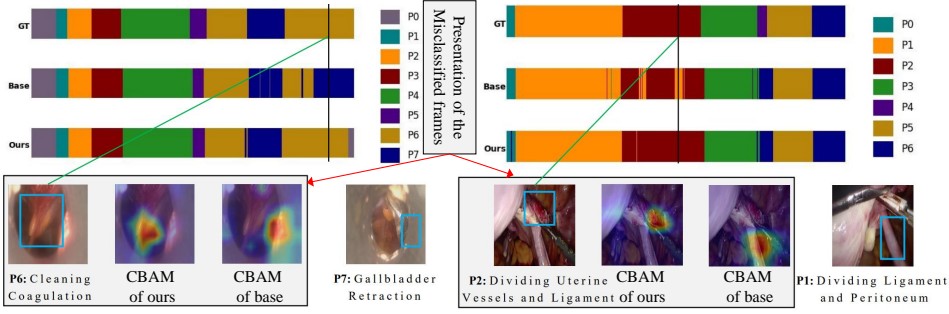

Figure 7: Visualizations presented on M2Cai16 (left) and AutoLaparo (right) datasets, where the blue box in the black box per dataset indicates the target organ and tool that should be focused on while the blue box outside the black box represents incorrectly focused area. Surgical phases are best viewed in color of ribbon diagrams, such as $P_0$ to $P_7$ in M2Cai16. Phase names can be found in Appendix A.3. We observe that imbalanced phase distribution occurs in the ribbon diagrams. Additionally, we find that low-quality frame image with high semantic similarity across different categories may raise ambiguity for precise recognition.

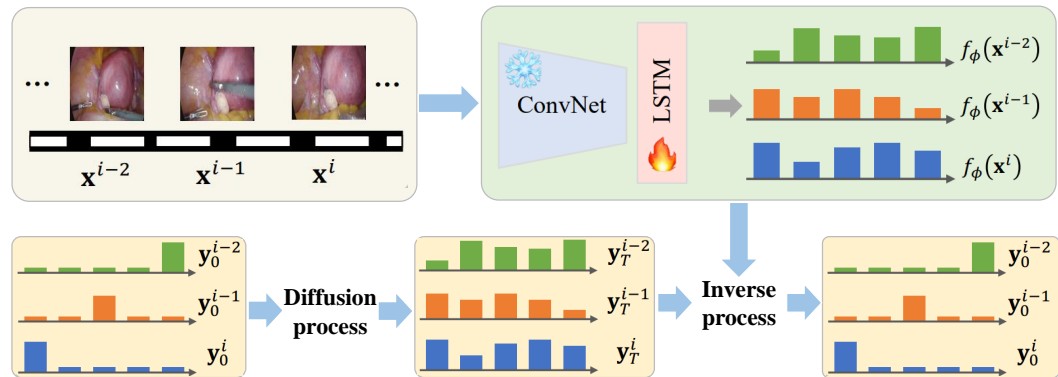

Figure 8: Overview of our MetaDiff consisting of a Classifier-guided Diffusion Model (CDM) and a Meta-weighted Optimization Algorithm (MOA). We employ a simple yet effective backbone $f_\phi$, ConvNext + LSTM, to deliver rough predictions. The upper part illustrates the data flow of obtaining rough predictions, while the lower part shows how CDM obtains clean prediction $y_0^i$ from rough prediction $f_\phi(x^i)$ for the $i$-th frame. The MOA is designed to train the CDM so that the surgical phase recognition could be robust against uncertainty.

Table 12: Results on the Cholec80 dataset. R denotes relaxed metric. The best results are marked in bold.

| Method | R | Cholec80 | | | |
| --- | --- | --- | --- | --- | --- |
| | | ACC | Precision | Recall | Jaccard |
| Online SurgPLAN++ | | 92.7 | 91.1 | 89.8 | 81.4 |
| Offline SurgPLAN++ | | 94.1 | **93.3** | 92.9 | 83.5 |
| SR-Mamba | ✓ | 92.6 | 90.3 | 90.6 | 81.5 |
| SPR-Mamba | ✓ | 93.1 | 89.3 | 90.1 | 81.4 |
| MetaDiff+ConvNext | | 94.2 | 89.6 | 90.0 | 81.7 |
| MetaDiff+ConvNext | ✓ | **95.3** | 92.9 | **93.1** | **86.0** |

Table 13: Results on the OphNet dataset using metrics following OphNet. The best results are marked in bold.

| Method | Acc-top1 | Acc-top5 | Params | inference time |
| --- | --- | --- | --- | --- |
| X-CLIP16 | 64.8 | 89.3 | 194.9M | 216ms |
| X-CLIP32 | 71.2 | 91.6 | 194.9M | 243ms |
| Ours | 69.7 | 91.4 | **32.8M** | **170**ms |
| Ours+X-CLIP32 | **72.1** | **92.4** | 194.9M | 256ms |

Table 14: Results on the Cholec80 dataset using different backbones. The best results are marked in bold.

| Method | R | Cholec80 | | | |
| --- | --- | --- | --- | --- | --- |
| | | ACC | Precision | Recall | Jaccard |
| MetaDiff+ResNet | ✓ | 94.8 | 90.7 | 92.8 | 84.5 |
| MetaDiff+ViT | ✓ | 95.0 | 91.7 | **93.2** | 85.4 |
| MetaDiff+ConvNext | ✓ | **95.3** | **92.9** | 93.1 | **86.0** |

Table 15: Results on the OphNet dataset regarding the effect of background frames.

| Method | Acc | Precision | Recall | Jaccard |
| --- | --- | --- | --- | --- |
| ignore bg | 73.0 | 63.2 | 56.3 | 55.6 |
| ignore bgl | 76.3 | 64.3 | 58.4 | 59.3 |
| use 1 bgl | 70.0 | 56.2 | 51.7 | 52.5 |

Table 16: Ablation studies on Train Time, CPU and GPU Memories during training.

| MOA | CDM | Train Time | CPU Mem | GPU Mem |
| --- | --- | --- | --- | --- |
| × | × | 19:34:52 | 4.93G | 7.99G |
| ✓ | × | 30:12:21 | 6.72G | 9.44G |
| × | ✓ | 25:58:19 | 8.18G | 9.37G |
| ✓ | ✓ | 34:35:16 | 10.22G | 10.82G |

Table 17: Results on the NurViD dataset using the metrics following NurViD. The best results are marked in bold.

| Method | Many(9) | Medium(66) | Few(87) | All(162) |
| --- | --- | --- | --- | --- |
| SlowFast | 29.8 | 15.5 | 7.9 | 21.1 |
| C3D | 28.1 | 14.6 | 7.3 | 22.8 |
| I3D | 31.3 | 14.8 | 8.2 | 21.5 |
| Ours | **34.2** | **22.5** | **19.3** | **26.0** |

