# OpenReview forum: "Meta-weighted Diffusion Model for Reliable Online Surgical Phase Recognition"
_ICLR.cc/2025/Conference — Submitted to ICLR 2025_

### Official Review · Reviewer_YAUo · 2024-10-31

**Soundness:** 4
**Presentation:** 3
**Contribution:** 3
**Rating:** 6
**Confidence:** 4

**Summary:**

This paper presents Meta-weighted Diffusion Model (MetaDiff) for reliable online surgical phase recognition. MetaDiff aims to enhance phase recognition accuracy by addressing uncertainties inherent in surgical videos, primarily due to imbalanced phase distribution and variable image quality. The model integrates a Classifier-guided Diffusion Model (CDM) to manage noise in predictions, and a Meta-weighted Optimization Algorithm (MOA) for reweighting, thus focusing on minority phases that might otherwise be underrepresented. Experimental evaluations on four surgical datasets—Cholec80, AutoLaparo, M2Cai16, and CATARACTS—demonstrate MetaDiff’s strong performance in terms of accuracy and uncertainty estimation, outperforming existing models in accuracy metrics.

**Strengths:**

(1) The work is innovative in combining meta-learning and diffusion models to tackle uncertainties in surgical video recognition, particularly focusing on imbalanced data issues and varying image quality.

(2) The paper is methodologically rigorous, with a thorough background in generative models, uncertainty quantification (e.g., Prediction Interval Width and Paired Two-Sample t-Test), and the use of meta-learning for handling imbalanced datasets. The proposed MetaDiff model and its evaluation are well-supported with extensive experiments on diverse surgical datasets, showing notable improvements over baseline methods.

**Weaknesses:**

(1) How does the classifier-guided diffusion process handle scenarios where phase boundaries are ambiguous, and does this affect MetaDiff’s stability across various surgical tasks?

(2) Could the meta-weight net introduce biases if specific phases dominate the meta dataset? If so, how is this mitigated?

(3) Given that uncertainty estimation was key to the model's objectives, would there be cases where the uncertainty metrics (PIW and PTST) lead to false negatives or positives?

(4) The author could try conducting experiments on a larger-scale and more fine-grained phase recognition dataset, such as OphNet
(https://github.com/minghu0830/OphNet-benchmark).

**Questions:**

Is there potential to generalize MetaDiff to other medical image tasks that face similar uncertainty issues, or are the diffusion model adjustments tailored specifically to the temporal structure of surgical videos?

---

> ### Author Response · Authors · 2024-11-23
>
> W1: (1) We propose a classifier-guided diffusion model to obtian reliable surgical phase prediction by regularizing uncertainties including imbalance phase distribution and low image quality in surgical videos. As for low image quality, in this work, it refers to frame image with high semantic similarity across different phase categories that may raise ambiguity for precise recognition, which is exactly reviewer YAUo concerned. at training time, given an input surgical video frame, our model can output multiple prediction trajectories for the input frame. By calculating the expectation in Eq. 9, we aggregate information from these trajectories and ensure that the distribution of denoised predictions for a given training video frame is more compact around its true label, thus it helps us to regulate uncertainty to some extent. At test time, we can also utilize these multiple trajectories for each test video frame, using PIW and t-tests to evaluate the uncertainty of the denoised predictions, as described in Section 2.3. Intuitively, if the variance of the denoised predictions for a given test video frame is high, the uncertainty for that frame is also likely high, which can be used to remind users to get noticed. (2) As for evaluation, we also employ relaxed metric when unclear boundaries occur. The relaxed metric allows frames within a certain time range at the intersection of two phases to be classified as belonging to either phase and still be considered correct.
>
> W2: In order to assign appropriate weights for different video frames in the case of imbalanced phase distribution, we design a meta learning strategy to optimize ELBO in Eq. 9 based on a balanced meta dataset. Namely, different phases dominate the meta dataset equally. Hence, there should be no concern about the biases for meta-weight net caused by imbalanced meta dataset.
>
> W3: (1) As discussed in W1 (1) for reviewer YAUo, uncertainty regularization for model parameters is already been implemented at training time. That is, given some ambiguous video frames, the model can try it best to give predictions with low variance. (2) As for evaluation, PIW ensures that the model’s predictions cover the range of the true values by providing a reasonable prediction interval, preventing excessive contraction and thereby reducing the probability of false negatives. Meanwhile, the PTST metric measures the effectiveness of the prediction interval in containing the true value, ensuring that the prediction interval correctly covers the target in most cases, thus reducing the probability of false positives. These two metrics work together to balance the width of the prediction interval and the accuracy of covering the true value, statistically lowering the occurrence of false negative and false positive events.
>
> W4: We conduct experiments on the OphNet dataset and replenish results in Table 13 of the appendix A6. As we can see that our model outperforms other competitive competitors. For convienence, we show Table 13 as follows:
>
> **Table 13** : Results on the OphNet dataset using metrics following OphNet. The best results are marked in bold.
>
> | Method          | Acc-top1 | Acc-top5 |
> |-----------------|----------|----------|
> | I3D             | 30.2     | 71.2     |
> | SlowFast        | 31.7     | 61.8     |
> | X3D             | 33.5     | 63.2     |
> | MViT V2         | 28.3     | 60.2     |
> | x-CLIP32        | 62.7     | 85.8     |
> | VIFI-CLIP32     | 68.4     | 87.2     |
> | Ours       | **69.7** | **91.4** |
>
> Q1: (1) In this work, we focus on the task of surgical videos, hence, we employ a simple backbone consists of ConvNext and LSTM to merge temporal features for constructing conditions of MetaDiff; (2) MetaDiff is flexible and could also be generalize to other medical image tasks involving uncertainty issues with only a few modification, that is, we only need to replace backbones for videos with those for images.

---

> > ### Author Response · Authors · 2024-11-23
> >
> > Dear Reviewer YAUo,
> >
> > We appreciate it if you could let us know whether our responses are able to address your concerns. We're happy to address any further concerns. Thank you,
> >
> > Best wishes.
> >
> > Paper 8858 Authors

---

> > > ### Comment · Reviewer_YAUo · 2024-11-23
> > >
> > > Thank you for your efforts in conducting the supplementary experiments. In fact, I’ve recently been working with the OphNet dataset as well. However, I’ve noticed that the dataset in the open-source version provided by the authors seems to differ from the version they used in their ECCV paper (possibly due to adjustments made for an upcoming challenge). Additionally, it seems that the baseline results for their new version of the dataset have not been released yet. Could this cause some issues with the comparison of results? If you could clarify this matter, I would be more than happy to improve my rating.

---

> > > > ### Author Response · Authors · 2024-11-24
> > > >
> > > > Dear reviewer YAUo,
> > > >
> > > > Thanks for your suggestions and reply. We are the first time to deal with the OphNet dataset, as far as we know, the dataset was released several days ago. However, we do not notice that the released dataset version is different from their ECCV papers' and just report results from their paper. And yes, there may occur unfairness. Thanks to your kind remind, we are conducting experiments on the released OphNet dataset using the baselines compared in their ECCV paper like X-CLIP32 and ViFi-CLIP32, both of them have official codes. On the other hand, we have contacted the authors of ECCV paper and request them to give us the annotated CSV files so that we could process another OphNet dataset consistent with their paper, then we can directly use the reported results in their paper. To sum up, we will replenish updated results as soon as possible. Thanks for your patience again.

---

> > > > ### Author Response · Authors · 2024-11-25
> > > >
> > > > Dear reviewer YAUo,
> > > >
> > > > We have finished experiments on the released OphNet dataset using official codes of baselines (X-CLIP16 and X-CLIP32) reported in their ECCV paper. Additionally, we also replace backbone of ConvNext+LSTM with X-CLIP32 and find consistent performance improvements. Results are reported in Table 13 of the appendix A6. For convienence, we report Table 13 as follows:
> > > >
> > > > **Table 13**: Results on the OphNet dataset using metrics following OphNet. The best results are marked in bold.
> > > >
> > > > | Methods | Top1 | Top5 | Params | ISPSoA | Pre-trained Dataset |
> > > > |:----:|:----:|:----:|:----:|:----:|:----:|
> > > > | X-CLIP16 | 64.8 | 89.3 | 194.9M | 216ms | CLIP-400M |
> > > > | X-CLIP32 | 71.2 | 91.6 | 194.9M | 243ms | CLIP-400M |
> > > > | ours | 69.7 | 91.4 | **32.8M** | **170ms** | imageNet-21K |
> > > > | ours+X-CLIP32 | **72.1** | **92.4** | 194.9M | 256ms | CLIP-400M |

---

> > > > > ### Comment · Reviewer_YAUo · 2024-11-25
> > > > >
> > > > > Thank you to the authors for the clarification and the additional experiments, which addressed my concerns. As a result, I have raised my rating.

---

> > > > > > ### Author Response · Authors · 2024-11-25
> > > > > >
> > > > > > Thanks for your reply, we are glad to address any other unsolved concerns to further improve the quality of this work with our common efforts.

---

### Official Review · Reviewer_4qJC · 2024-11-02

**Soundness:** 2
**Presentation:** 3
**Contribution:** 2
**Rating:** 6
**Confidence:** 3

**Summary:**

This paper identifies the unsolved problems of uncertainty caused by low-quality images and imbalanced phase distribution in surgical phase recognition. It introduces a meta-weighted diffusion model (MetaDiff) as a solution. MetaDiff integrates a classifier-guided diffusion model (CDM) that conditions the diffusion process on class predictions. Additionally, it employs a meta-weighted objective function (MOA) during optimization to learn weights that rebalance the training loss, effectively mitigating the impact of imbalanced phase distribution throughout training. Extensive experiments across four benchmarks demonstrate the promising performance of this approach.

**Strengths:**

* While previous works focus on learning strong and long-term spatial-temporal representations, this paper is the first to identify the uncertainty problem in surgical phase recognition.This challenge could have significant implications for deep learning applications in surgical settings.
* The application of meta-learning to rebalance class weights is an elegant approach.
* The results show that the proposed approach effectively improves the performance without increasing the training time.
* Experiments are extensive.

**Weaknesses:**

The paper's writing is often unclear, and clarity needs improvement to enhance readability and comprehension.

Some confusion and suggestions for the writing:
* In lines 38 and 39, mentioning the objective in the introduction in a mathematical format without defining the variables ($L$, $C$, $I$) is confusing.
* Figure titles should only describe the corresponding figure, while the objective is included in the title of Figure 1, which is irrelevant to the figure content. Additionally, the two images in the black solid boxes of each example are not explained.
* citation format makes the text a bit hard to read. It would be good if the authors could check the ICLR citation format and make updates to the paper.
* usages of notations are inconsistent, e.g., i in Eq. (4) and $i$ in the text (line 141).
* multiple grammar errors, e.g., line 157 " CDM is consisted of a ..." -> " CDM consists of ..."
* line 160-161 "Notably, ... which are **modestly collected** by the MOA used to learn parameters of CDM" is confusing. What does "**modestly**" mean in this context?
* line 207-208 "we use model the diffusion process endpoint through the incorporation of the conditional embedding.. " is unclear.
* in Figure 2, CDM and MOA should be illustrated clearly (put CDM's learning process in one box potentially?), and in the prediction process, replacing the DM component with "......" makes the whole process inconsistent. Also, the CDM learning process should be illustrated more clearly.
* in Section 3.2, video frames and clips are used interchangeably, leading to minor confusion.
* in Table 3, LoViT (Pr) on AutLapro is the best while not bolded, and the corresponding description in lines 354-355 is not accurate.
* Table 6 title "DM" -> "CDM"

**Questions:**

* I don't see how the CDM addresses the uncertainty introduced by image quality. It appears that the conditioned diffusion process primarily aligns the class inference with the provided ground truth labels. Could you elaborate on this aspect further?
* Could you provide details on the training time and memory usage? Will the MOA and CDM significantly increase computation time?

---

> ### Author Response · Authors · 2024-11-23
>
> W: We explain the writing confusions as follows and will make them clear in the final version of manuscript.
>
> (1) L, C, I in line 38 and line 39 refer to video frame length, category of phase number, and channel number per frame, respectively.
>
> (2) We amend Fig.1 of the manuscript to Fig. 7 of the appendix A6 and try to make it clearer.
>
> (3) We have used the ICLR format citation in our template and the local PDF in our computer is normal with green box citation, however, once we update it online, the green box disappears. Fortunately, we check that the hyperlink feature is still there. We will fix that in the final version of manuscript.
>
> (4, 5) We modify the notation inconsistencies and grammer errors and make more careful proofreading.
>
> (6) The meaning of "modestly" in the context: In order to obtain reliable predictions by awaring the uncertainties conveyed in surgical video frames, rather than calculate one prediction per frame, our model outputs multiple prediction trajectories per frame. At training and test time, we should collect countable amount of predictions per frame to carry out model learning and inference, respectively. And we refer this behavior as "modest collection".
>
> (7) We modify the expression ""we use model the diffusion process endpoint through the incorporation of the conditional embedding.. "" to "we model the diffusion process endpoint ..." by deleting "use".
>
> (8) We amend Fig.2 of the manuscript to Fig. 8 of the appendix A6 and try to make it clearer. Meanwhile, CDM is our proposed parametric diffusion model while MOA is the developed algorithm to opimize parameters of CDM. Combining CDM and MOA forms  our MetaDiff model.
>
> (9) We replace all "clip" in Sec. 3.2 with "frame" for clarity.
>
> (10) We bold LoViT (Pr) and modify the corresponding description as: "MetaDiff significantly outperforms most of competitors, such as SKiT and LAST, across various metrics including accuracy (Acc), precision (Pr), recall (Re), and Jaccard (Ja)."
>
> (11) We change "DM" to "CDM" in Table 6 of the manuscript.
>
> Q1: (1) As illustrated in Fig. 7 of the appendix A6 (a modified version of Fig. 1), image quality in this work mainly refers to frame images with high semantic similarity across different phase categories that may raise ambiguity for precise recognition. (2) We propose a classifier-guided diffusion model to deal with uncertainty in surgical phase recognition mainly caused by image quality by leveraging the model’s generative capability. Specifically, at training time, given an input surgical video frame, our model can output multiple prediction trajectories for the input frame. By calculating the expectation in Eq. 9, we aggregate information from these trajectories and ensure that the distribution of denoised predictions for a given training video frame is more compact around its true label, thus it helps us to regulate uncertainty to some extent. At test time, we can also utilize these multiple trajectories for each test video frame, using PIW and t-tests to evaluate the uncertainty of the denoised predictions, as described in Section 2.3. Intuitively, if the variance of the denoised predictions for a given test video frame is high, the uncertainty for that frame is also likely high, which can be used to remind users to get noticed.
>
> Q2: We replenish details on the training time and memory usage in Table 16 of the appendix A6. As for inference, results have been reported in Table 5 of the manuscript. For convienence, we report Table 16 as follows:
>
> **Table 16. Ablation studies on training time, CPU and GPU memories during training.**
>
> | MOA  | CDM  | Train Time | CPU Mem | GPU Mem |
> |------|------|------------|---------|---------|
> | **✘** | **✘** | 19:34:52   | 4.93G   | 7.99G   |
> | **✔** | **✘** | 30:12:21   | 6.72G   | 9.44G   |
> | **✘** | **✔** | 25:58:19   | 8.18G   | 9.37G   |
> | **✔** | **✔** | 34:35:16   | 10.22G  | 10.82G  |

---

> > ### Author Response · Authors · 2024-11-23
> >
> > Dear Reviewer 4qJC,
> >
> > We appreciate it if you could let us know whether our responses are able to address your concerns. We're happy to address any further concerns. Thank you,
> >
> > Best wishes.
> >
> > Paper 8858 Authors

---

> > > ### Author Response · Authors · 2024-11-25
> > >
> > > Dear Reviewer 4qJC,
> > >
> > > Given that the rebuttal phase is about to end, we would like to know if our responses have addressed your concerns. Additionally, we are still here and happy to address any further concerns.
> > >
> > > Best wishes,
> > >
> > > Paper 8858 Authors

---

> > > > ### Comment · Reviewer_4qJC · 2024-11-25
> > > >
> > > > Dear Authors,
> > > >
> > > > Thank you for providing a detailed clarification of your method. I appreciate your effort in addressing my concerns, particularly regarding the writing and computational cost. I am happy to improve my original score.

---

> > > > > ### Author Response · Authors · 2024-11-27
> > > > >
> > > > > Dear reviewer 4qJC,
> > > > >
> > > > > Thanks for your suggestions and reply, and we are glad to address any other unsolved concerns to further improve the quality of this work with our common efforts.
> > > > >
> > > > > Best wishes.
> > > > >
> > > > > Paper 8858 Authors

---

### Official Review · Reviewer_EE1r · 2024-11-02

**Soundness:** 3
**Presentation:** 4
**Contribution:** 3
**Rating:** 5
**Confidence:** 5

**Summary:**

This paper addresses the challenge of uncertainty in surgical phase recognition by proposing a meta-weighted diffusion model (MetaDiff) that leverages meta-learning and classifier-guided diffusion for improved reliability. The model is designed to address issues caused by imbalanced phase distribution and low-quality image acquisition, common in surgical videos, which can lead to unreliable phase recognition outcomes. MetaDiff combines a classifier-guided diffusion model (CDM) with a meta-weighted optimization algorithm (MOA) to manage these uncertainties, presenting significant improvements in accuracy over previous methods on benchmark datasets, including Cholec80, AutoLaparo, M2Cai16, and CATARACTS.

**Strengths:**

1.	The integration of a diffusion model (specifically, a classifier-guided diffusion model) for handling uncertainty in predictions is innovative. By generating multiple denoised trajectories for each frame, the model can assess the confidence of predictions based on statistical measures like Prediction Interval Width (PIW) and the Paired Two-Sample t-Test (PTST). This enables more robust decision-making, especially in high-stakes scenarios like surgical phase recognition where reliability is critical.
2.	The use of a meta-weighted optimization algorithm (MOA) that dynamically re-weights the loss function for each video frame is an effective strategy to counteract data imbalance across phases.
3.	The paper uses a carefully chosen ConvNeXt backbone paired with LSTM to capture long-range dependencies in surgical video frames without leaking future information, crucial for real-time phase recognition. This pairing is technically sound as ConvNeXt focuses on spatial features, while LSTM efficiently captures temporal dynamics.

**Weaknesses:**

1.	The performance of MetaDiff is partially dependent on ConvNeXt+LSTM pre-training. For tasks beyond surgical videos, this dependence could limit the model’s adaptability, especially if ConvNeXt features do not generalize well.
2.	Although MetaDiff is well-evaluated on four datasets, these datasets cover similar surgical contexts. The model’s robustness to highly variable surgical environments (e.g., laparoscopic vs. robotic vs. open surgery) remains untested.
3.	High dependency on accurate timestamp annotations. Surgical video data often contains inaccurate timestamp annotations, especially when manually labeled. And underexplored sensitivity to non-surgical contextual information, some frames may contain non-surgical context (e.g., camera adjustments or the organ scene is shown, but no procedure is performed).
4.	Some new but not necessarily SOTA works could also be included for comparison, such as：
[1] SurgPLAN++: Universal Surgical Phase Localization Network for Online and Offline Inference. https://arxiv.org/pdf/2409.12467
[2] SR-Mamba: Effective Surgical Phase Recognition with State Space Model. https://arxiv.org/pdf/2407.08333
[3] SPRMamba: Surgical Phase Recognition for Endoscopic Submucosal Dissection with Mamba. https://arxiv.org/abs/2409.12108

**Questions:**

please check the Weakness section

---

> ### Author Response · Authors · 2024-11-23
>
> W1: (1) Performance of most of the downstream tasks depend on abilities of pre-trained backbones, and our model in the scope of surgical phase recognition is also no exception. For tasks that ConvNext do not generalize well, if we still want to use our model, other domain adaptation techniques for ConvNext such as fine-tuning need to be considered. (2) We replenish other widely used pre-trained backbones including ResNet and ViT to validate robustness of our model, results are reported in Table 14 of the appendix A6. Again, we claim that such kind of robustness comes from our well-designed conditional diffusion model for modeling uncertainties in surgical videos. For convenience, we also show Table 14 as follows:
>
> **Table 14**: Results on the Cholec80 dataset using different backbones. The best results are in bold.
>
> | backbone | Acc | Pr | Re | Ja |
> | :---: | :---: | :---: | :---: | :---: |
> | MetaDiff+ResNet| 94.8 | 90.7 | 92.8 | 84.5 |
> | MetaDiff+ViT| 95.0 | 91.7 | **93.2** | 85.4 |
> | MetaDiff+ConvNext | **95.3** | **92.9** | 93.1 | **86.0** |
>
> W2: We validate our model on an nursing procedure video dataset NurViD [1] and report results in Table 17 of the appendix A6. The results demonstrate that our model also achieves promising performance compared with other competitors thanks to our well-designed uncertainty-awared conditional diffusion model.
>
> [1] NurViD: A Large Expert-Level Video Database for Nursing Procedure Activity Understanding
>
> W3: (1) We have conducted experiments on some practical and challenge situations including semi-supervised and few-shot surgical phase recognition tasks, results have been reported in Fig. 5(a) of the manuscript. As for semi-supervised setting, we downsample the whole video with ratio ranging from 0.2 to 1, and use both the sampled labeled frames and non-sampled unlabeled frames (without labels) to train our model. As for few-shot setting, we only use the sampled labeled frames to train our model. (2) Annotations for surgical phase recognition are inherently coarse-grained and manually labeled. For cases where boundary ambiguity arises due to manual labeling, the paper addresses this issue using the relaxed metric. The relaxed metric allows frames within a certain time range at the intersection of two phases to be classified as belonging to either phase and still be considered correct. (3) When it comes to background frames for non-surgical phases like camera adjustments and so on. We implement experiments on the OphNet dataset as follows: i) ignoring the background frames (ignore bg); ii) ignoring the background labels while using their frames (ignore bgl), and iii) treating the background frames as a single class (use 1 bgl).  Results are shown in Table 15 of the appendix A6. For convienence, we report Table 15 as follows:
>
> **Table 15**: Results on the OphNet dataset regarding the effect of background frames.
>
> | Method       | Acc   | Precision | Recall | Jaccard |
> |--------------|-------|-----------|--------|---------|
> | ignore bg    | 73.0  | 63.2      | 56.3   | 55.6    |
> | ignore bgl   | 76.3 | 64.3  | 58.4 | 59.3 |
> | use 1 bgl    | 70.0  | 56.2      | 51.7   | 52.5    |
>
> W4: We replenish comparisons between our model and reviewer suggested methods including SurgPLAN++, SR-Mamba, and SPR-Mamba in Table 12 of the appendix A6. Our model achieves competitive or even better results compared with the competitors. For convienence, we show Table 14 as follows:
>
> **Table 14**: Results on the Cholec80 dataset. R denotes the relaxed metric. The best results are marked in bold.
>
> | Method                | R   | ACC   | Precision | Recall | Jaccard |
> |-----------------------|-----|-------|-----------|--------|---------|
> | Online SurgPLAN++     |     | 92.7  | 91.1      | 89.8   | 81.4    |
> | Offline SurgPLAN++    |     | 94.1  | **93.3**  | 92.9   | 83.5    |
> | SR-Mamba              | ✓   | 92.6  | 90.3      | 90.6   | 81.5    |
> | SPR-Mamba             | ✓   | 93.1  | 89.3      | 90.1   | 81.4    |
> | MetaDiff+ConvNext     |     | 94.2  | 89.6      | 90.0   | 81.7    |
> | MetaDiff+ConvNext     | ✓   | **95.3** | 92.9  | **93.1**   | **86.0** |

---

> > ### Author Response · Authors · 2024-11-23
> >
> > Dear Reviewer EE1r,
> >
> > We appreciate it if you could let us know whether our responses are able to address your concerns. We're happy to address any further concerns. Thank you,
> >
> > Best wishes.
> >
> > Paper 8858 Authors

---

> > > ### Author Response · Authors · 2024-11-25
> > >
> > > Dear Reviewer EE1r,
> > >
> > > Given that the rebuttal phase is about to end, we would like to know if our responses have addressed your concerns. Additionally, we are still here and happy to address any further concerns.
> > >
> > > Best wishes,
> > >
> > > Paper 8858 Authors

---

> > > ### Author Response · Authors · 2024-11-27
> > >
> > > Dear Reviewer EE1r,
> > >
> > > We appreciate it if you could let us know whether our responses are able to address your concerns. We're happy to address any further concerns. Thank you,
> > >
> > > Best wishes.
> > >
> > > Paper 8858 Authors

---

> > > > ### Author Response · Authors · 2024-11-28
> > > >
> > > > Dear Reviewer EE1r,
> > > >
> > > > We appreciate it if you could let us know whether our responses are able to address your concerns. We're happy to address any further concerns. Thank you,
> > > >
> > > > Best wishes.
> > > >
> > > > Paper 8858 Authors

---

> > > > > ### Author Response · Authors · 2024-11-30
> > > > >
> > > > > Dear Reviewer EE1r,
> > > > >
> > > > > Given that the rebuttal phase is about to end, we would like to know if our responses have addressed your concerns. Additionally, we are still here and happy to address any further concerns.
> > > > >
> > > > > Best wishes,
> > > > >
> > > > > Paper 8858 Authors

---

> > > > > > ### Author Response · Authors · 2024-12-01
> > > > > >
> > > > > > Dear Reviewer EE1r,
> > > > > >
> > > > > > Given that the rebuttal phase is about to end, we would like to know if our responses have addressed your concerns. Additionally, we are still here and happy to address any further concerns.
> > > > > >
> > > > > > Best wishes,
> > > > > >
> > > > > > Paper 8858 Authors

---

### Official Review · Reviewer_jaDG · 2024-11-04

**Soundness:** 3
**Presentation:** 2
**Contribution:** 3
**Rating:** 6
**Confidence:** 2

**Summary:**

The paper addresses a highly relevant problem in surgical phase recognition, emphasizing its importance due to the direct impact on human life and health. This underscores the paper's significance and the potential real-world impact of the proposed solutions. The authors effectively highlight two key sources of uncertainty—imbalanced phase distribution and low-quality image acquisition—which are critical but often overlooked issues in surgical video analysis. Recognizing these challenges and focusing on their mitigation shows a deep understanding of the domain's requirements. The introduction of a meta-weighted diffusion model (MetaDiff) is an innovative approach.

**Strengths:**

The use of meta-learning combined with a diffusion model to address uncertainties is unique and relevant, particularly for medical applications where reliability is crucial. The paper provides an extensive evaluation on four benchmark datasets (Cholec80, AutoLaparo, M2Cai16, and CATARACTS). This demonstrates the generalizability and robustness of the proposed model across a diverse range of surgical procedures, enhancing the credibility of the results. Experimental results indicate that MetaDiff outperforms state-of-the-art methods, achieving impressive accuracy improvements across all tested datasets. This supports the effectiveness of the proposed method and its potential for real-world applications.

**Weaknesses:**

1. The classifier-guided diffusion model is a core part of the approach, yet its description lacks sufficient technical depth. More explanation on how it specifically handles uncertainty caused by image quality, along with any potential limitations, would make the model easier to understand and evaluate.
2. Although the model performs well on surgical video benchmarks, there’s limited discussion on its adaptability to other medical video analysis tasks (e.g., surgical process planning) or datasets with differing visual and temporal characteristics. Expanding on this aspect could strengthen the broader applicability of MetaDiff.

**Questions:**

1.Guiding diffusion model training with classification is not a new approach; for example, in "Pdpp: Projected Diffusion for Procedure Planning in Instructional Videos and See" and "Predict, Plan: Diffusion for Procedure Planning in Robotic Surgical Videos", task and phase class information is used to guide planning task learning via diffusion models. Are there similarities or differences in using classifiers as guides for diffusion models between phase recognition tasks and planning tasks?
2. In Figure 2, the input consists of three consecutive frames. Are there significant differences among these frames? Given the typically long annotated duration for each phase, the visual information across different sets of three frames should vary considerably. Is there a mechanism to assess this variability?

---

> ### Author Response · Authors · 2024-11-23
>
> W1: We propose a classifier-guided diffusion model to deal with uncertainty in surgical phase recognition mainly caused by image quality by leveraging the model’s generative capability. Specifically, at training time, given an input surgical video frame, our model can output multiple prediction trajectories for the input frame. By calculating the expectation in Eq. 9, we aggregate information from these trajectories and ensure that the distribution of denoised predictions for a given training video frame is more compact around its true label, thus it helps us to regulate uncertainty to some extent. At test time, we can also utilize these multiple trajectories for each test video frame, using PIW and t-tests to evaluate the uncertainty of the denoised predictions, as described in Section 2.3. Intuitively, if the variance of the denoised predictions for a given test video frame is high, the uncertainty for that frame is also likely high, which can be used to remind users to get noticed.
>
> W2: We further validate the broader applicability of MetaDiff on ophthalmic surgical videos from OphNet [1] and  nursing procedure videos from NurViD[2] and report results in Table 13 and Table 17 of the appendix A6. As we can see that our model achieves promising performance compared with the most recently developed competitors. For convenience, we report Table 13 and Table 17 as follows:
>
> **Table 13**: Results on the OphNet dataset using metrics following OphNet.The best results are marked in bold.
>
> | Methods | top1 | top5 |
> | :---: | :---: | :---: |
> | I3D | 30.2 | 71.2 |
> | SlowFast | 31.7 | 61.8 |
> | X3D | 33.5 | 63.2 |
> | MViT V2 | 28.3 | 60.2 |
> | X-CLIP32 | 62.7 | 85.8 |
> | ViFi-CLIP32 | 68.4 | 87.2 |
> | MetaDiff(Ours) | **69.7** | **91.4** |
>
> **Table 17**: Results on the NurViD dataset using the metrics following NurViD. The best results are marked in bold.
>
> | Methods | Many(9) | Medium(66) | Few(87) | All(162) |
> | :---: | :---: | :---: | :---: | :---: |
> | SlowFast | 29.8 | 15.5 | 7.9 | 21.1 |
> | C3D | 28.1 | 14.6 | 7.3 | 22.8 |
> | I3D | 31.3 | 14.8 | 8.2 | 21.5 |
> | MetaDiff(Ours) | **34.2** | **22.5** | **19.3** | **26.0** |
>
>
> [1] OphNet: A Large-Scale Video Benchmark for Ophthalmic Surgical Workflow Understanding
>
> [2] NurViD: A Large Expert-Level Video Database for Nursing Procedure Activity Understanding
>
> Q1: (1) The tasks of PDPP, MS-PCD and ours are different, where the former two are video procedure planning and the latter is surgical phase recognition. As for PDPP and MS-PCD, the initial and end frames are concatenated along with Gaussian noise as condition for diffusion decoder to generate the intermediate frames with one trajectory per frame. On the contrary, our model takes features of the current surgical video frame as condition for decoding multiple prediction trajectories for that frame, which could be collected for reducing and evaluating uncertainty at training and test time, respectively. (2) Rather than intuitively concatenated conditions with Gaussian noise as in PDPP and MS-PCD do, we first model the condition as a Gaussian distrbution at the endpoint of forward process and derive a tractable backward process posterior corresponding to the forward process. Then we introduce an ELBO as in Eq. 9 of the manuscript to optimize parameters of our model.
>
> Q2: (1) As reviewer pointed out that the visual information across different sets of frames may vary considerably, in this work, we mainly use cascaded ConvNext and LSTM to fuse temporal information into the current frame feature so as to set the model aware of such frame variance. (2) Other temporal modeling techniques could also be considered, such as Transformer and its complicated variants. However, in this work, we mainly want to illustrate that modeling uncertainty with our well-designed condtional diffusion model could indeed bring some significant benefits, thus, we only employ a very basic backbone consisting of ConvNext and LSTM for computing conditions.

---

> > ### Author Response · Authors · 2024-11-23
> >
> > Dear Reviewer jaDG,
> >
> > We appreciate it if you could let us know whether our responses are able to address your concerns. We're happy to address any further concerns. Thank you,
> >
> > Best wishes.
> >
> > Paper 8858 Authors

---

> > > ### Comment · Reviewer_jaDG · 2024-11-23
> > >
> > > Thank you for the authors' response. Most of my concerns have been addressed, and I am therefore increasing my score to 6.

---

> > > > ### Author Response · Authors · 2024-11-23
> > > >
> > > > Thanks for your reply, we are glad to address any other unsolved concerns to further improve the quality of this work with our common efforts.

---

### Author Response · Authors · 2024-11-23

Dear reviewers and AC

Thanks a lot for your effort in reviewing this submission! We have tried our best to address the mentioned concerns/problems in the rebuttal. Feel free to let us know if there is anything unclear or so. We are happy to clarify them.

Best, Authors

---

### Author Response · Authors · 2024-11-25

Dear AC,

Given that the rebuttal phase is about to end, however, it seems that reviewers EE1r and 4qJC haven't read our responses. Suggestions of reviewers EE1r and 4qJC really do help us to improve the quality of this work, and we wonder that whether our responses addressed their concerns. Could you please remind the reviewers to check out our responses.

Best wishes.

Paper 8858 Authors

---

> ### Author Response · Authors · 2024-12-01
>
> Dear AC,
>
> Given that the rebuttal phase is about to end, however, it seems that reviewer EE1r hasn't read our responses. Could you please remind the reviewers to check out our responses and we wonder that whether our responses have addressed reviewer EE1r's concerns.
>
> Best wishes.
>
> Paper 8858 Authors

---

### Meta-Review · Area_Chair_cxpm · 2024-12-19

**Metareview:**

This paper presents an approach for improving surgical phase recognition. The authors mainly combines several existing algorithms such as classifier guided diffusion with uncertainty. While such combination could be new for this particular application, there is many other methods that leveraging on uncertainty to improve accuracy. To me, this is not a very significant contribution.

There are also some issues in the presentation. For example, classifier guided diffusion is not new while the authors did not make this very clear in their writing. The authors shall also review related work on uncertainty for various tasks and explain why this particular method is adopted for uncertainty estimation.

Overall, although this paper receives a relatively higher score among the border papers, I noticed that one senior reviewer ranked the paper below acceptance level with high confidence. This reviewer also raised a few concerns on the comparison with SOTA, the model’s robustness to highly variable surgical environments, etc.

Given these weakness, I am sorry that I cannot recommend to accept this paper. I would suggest the reviewers to take these comments into consideration and submit to next venue. PS: there are also some typos and formatting issues that need to be fixed.

**Additional Comments On Reviewer Discussion:**

NA

---

### Decision · Program_Chairs · 2025-01-22

Reject